# Optimizing Adaptive Attacks against Watermarks for Language Models

**Abdulrahman Diaa**[1]  **Toluwani Aremu**[2]  **Nils Lukas**[2]

## Abstract

Large Language Models (LLMs) can be misused to spread unwanted content at scale. Content watermarking deters misuse by hiding messages in content, enabling its detection using a secret *watermarking key*. Robustness is a core security property, stating that evading detection requires (significant) degradation of the content's quality. Many LLM watermarking methods have been proposed, but robustness is tested only against *non-adaptive* attackers who lack knowledge of the watermarking method and can find only suboptimal attacks. We formulate watermark robustness as an objective function and use preference-based optimization to tune *adaptive* attacks against the specific watermarking method. Our evaluation shows that (i) adaptive attacks evade detection against all surveyed watermarks, (ii) training against *any* watermark succeeds in evading unseen watermarks, and (iii) optimization-based attacks are cost-effective. Our findings underscore the need to test robustness against adaptively tuned attacks. We release our adaptively tuned paraphrasers at `https://github.com/nilslukas/ada-wm-evasion`.

## 1. Introduction

A few Large Language Model (LLM) providers empower many users to generate human-quality text at scale, raising concerns about dual use (Barrett et al., 2023). Untrustworthy users can *misuse* the provided LLMs to generate harmful content, such as online spam (Weidinger et al., 2021), misinformation (Chen & Shu, 2024), or to facilitate phishing attacks (Shoaib et al., 2023). The ability to detect generated text can control these risks (Grinbaum & Adomaitis, 2022).

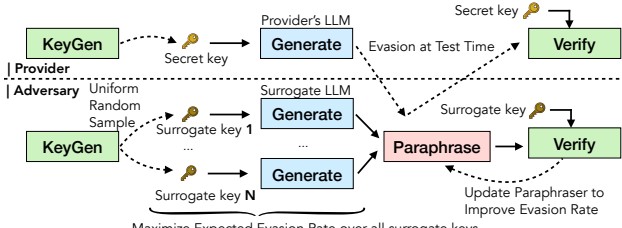

Figure 1: Adaptive attackers know the watermarking algorithms (KEYGEN, VERIFY), but not the secret key, so they can optimize a paraphraser against a specific watermark.

Content watermarking enables the detection of generated outputs by embedding hidden messages that can be extracted with a secret watermarking key. Some LLM providers, such as DeepMind (2024) and Meta (San Roman et al., 2024), have already deployed watermarking to promote the ethical use of their models. A threat to these providers are users who perturb generated text to evade watermark detection while preserving text quality. Such undetectable, generated text could further erode trust in the authenticity of digital media (Federal Register, 2023).

A core security property of watermarking is *robustness*, which requires that evading detection is only possible by significantly degrading text quality. Testing robustness requires identifying the most effective attack against a specific watermarking method. However, existing content watermarks for LLMs (Kirchenbauer et al., 2023a; Aaronson & Kirchner, 2023; Christ et al., 2023; Kuditipudi et al., 2024) test robustness only against *non-adaptive* attackers, who lack knowledge of the watermarking algorithms. This reliance on obscurity makes watermarking vulnerable to *adaptive* attacks (Lukas et al., 2024; Jovanović et al., 2024) when information about the watermarking algorithms is leaked.

We propose a method to curate preference datasets and adaptively optimize an attack against *known* content watermarking algorithms. Optimization is challenging due to (i) the complexity of optimizing within the discrete textual domain and (ii) the limited computational resources available to attackers. We demonstrate that adaptively tuned, open-weight LLMs such as `Llama2-7b` (Touvron et al., 2023) evade detection at negligible impact on text quality against `Llama3.1-70b` (Dubey et al., 2024). Our attacker spends

[1]David R. Cheriton School of Computer Science, University of Waterloo, Ontario, Canada [2]Mohammed Bin Zayed University of Artificial Intelligence (MBZUAI), Abu Dhabi, UAE. Correspondence to: Abdulrahman Diaa <abdulrahman.diaa@uwaterloo.ca>, Nils Lukas <nils.lukas@mbzuai.ac.ae>.

*Proceedings of the 42nd International Conference on Machine Learning*, Vancouver, Canada. PMLR 267, 2025. Copyright 2025 by the author(s).

less than 7 GPU hours to achieve an evasion rate of over 96% against any of the surveyed watermarking methods with negligible impact on text quality. Our attacks are Pareto optimal, even in the non-adaptive setting where they must transfer to unseen watermarks. Hence, future watermarking methods must consider our attacks to test robustness.

We make the following contributions. (1) We propose methods to curate preference-based datasets using LLMs, enabling us to adaptively fine-tune watermark evasion attacks against state-of-the-art language watermarks. (2) Adaptively tuned paraphrasers with 0.5-7 billion parameters evade detection from all tested watermarks at a negligible impact on text quality. We demonstrate their Pareto optimality for evasion rates greater than 90%[1]. Optimization against models with $46\times$ more parameters requires less than seven GPU hours, which challenges security assumptions, as even adversaries with limited resources can reliably evade detection using our attacks. (3) We test our attacks in the non-adaptive setting against unseen watermarks and demonstrate that they remain Pareto optimal compared to other non-adaptive attacks. Our results underscore the necessity of using optimizable, adaptive attacks to test robustness. (4) We publicly release our adaptively tuned paraphrasers to facilitate further research on robustness against adaptive attackers.

## 2. Background

**Large Language Models (LLMs)** estimate the probability distribution of the next token over a vocabulary $\mathcal{V}$ given a sequence of tokens. Autoregressive LLMs predict each subsequent token based on all preceding tokens. Formally, for a sequence of tokens $x_1, \ldots, x_n$, an LLM models:

$$P(x_n|x_1, \ldots, x_{n-1}) = \text{softmax}(f_\theta(x_1, \ldots, x_{n-1}))_n$$

where $f_\theta$ is a neural network with parameters $\theta$. Optimizing LLMs to maximize a reward function is challenging because the text is discrete, and the autoregressive generation process prevents direct backpropagation through the token sampling steps (Schulman et al., 2017).

**LLM Content Watermarking** hides a message in generated content that can later be extracted with access to the content using a secret watermarking key. A *watermarking method*, as formalized by (Lukas et al., 2024), comprises a set of algorithms (KEYGEN, EMBED, VERIFY):

- $\tau \leftarrow$ KEYGEN($\theta, \gamma$): A randomized function to generate a watermarking key $\tau$ given secret (i) LLM parameters $\theta$ and (ii) random seeds $\gamma \in \mathbb{R}$.

- $\theta^* \leftarrow$ EMBED($\theta, \tau, m$): Given a LLM $\theta$, a watermark-

ing key $\tau$ and a message $m$, this function[2] returns parameters $\theta^*$ of a *watermarked* LLM that generates watermarked text.

- $\eta \leftarrow$ VERIFY($x, \tau, m$): Detection involves (i) extracting a message $m'$ from text $x$ using $\tau$ and (ii) calculating the $p$-value $\eta$ for rejecting the null hypothesis that $m$ and $m'$ match by chance.

$(\epsilon, \delta)$**-Robustness.** A text watermark is a hidden signal in text that can be mapped to a message $m \in \mathcal{M}$ using a secret watermarking key $\tau$. The key $\tau$ refers to secret random bits of information used for detecting a watermark. A watermark is *retained* if VERIFY outputs $\eta < \rho$, for $\rho \in \mathbb{R}^+$. Let $Q : \mathcal{V}^* \times \mathcal{V}^* \to \mathbb{R}$ be a function to measure text quality between pairs of texts. We say that a watermark is $(\epsilon, \delta)$-robust if any paraphrase $y = \mathcal{A}(x)$ of a watermarked text $x$ that remains high-quality (i.e., $Q(x, y) > \delta$) also retains the watermark with probability $\geq 1 - \epsilon$. Let $\mathcal{A}$ be a randomized paraphrasing method, then robustness can be stated as follows.

$$\Pr_{y \leftarrow \mathcal{A}(x)} [\text{VERIFY}(y, \tau, m) \geq \rho \ \wedge \ Q(x, y) > \delta] < \epsilon \quad (1)$$

**Evasion Attacks.** Watermark evasion attacks are categorized by the attacker's access to the provider's (i) LLM, (ii) detection algorithm VERIFY that uses the provider's secret watermarking key, and (iii) knowledge of the watermarking algorithms. A *no-box* attacker has no access to the provider's LLM, whereas *black-box* attackers have API access, and *white-box* attackers know the parameters of the provider's LLM. *Online* attackers can query the provider's VERIFY functionality, as opposed to *offline* attackers who have no such access. *Adaptive* attackers know the algorithmic descriptions (KEYGEN, EMBED, VERIFY) of the provider's watermarking method, while *non-adaptive* attackers lack this knowledge. Our work focuses on no-box, offline attacks in adaptive and non-adaptive settings.

**Surveyed Watermarking Methods.** Following (Piet et al., 2023), we evaluate the robustness of four state-of-the-art watermarking methods[3] The `Exp` (Aaronson & Kirchner, 2023) method marks text by selecting tokens that maximize a score combining the conditional probability $P(x_n \mid x_0 \ldots x_{n-1})$ and a pseudorandom value derived from a sliding window of prior tokens. The `Dist-Shift` (Kirchenbauer et al., 2023a) method favours tokens from a green list, which is generated based on pseudorandom values and biases their logits to increase their selection probability. The `Binary` (Christ et al., 2023) approach converts tokens into bit-strings determined by pseudorandom values and

---

[1]Closed models such as GPT-4o are also on the Pareto front (due to high text quality) but achieve lower evasion rates.

[2]EMBED can modify the entire inference process.

[3]In Section A.4, we evaluate against more watermarks including SynthID (Dathathri et al., 2024), Unigram (Zhao et al., 2024) and SIR (Liu et al., 2024).

the language model's bit distribution, subsequently translating the bit-string back into a token sequence. Lastly, the `Inverse` (Kuditipudi et al., 2024) scheme uses inverse transform sampling by computing a cumulative distribution function ordered pseudorandomly according to a secret key and using a fixed pseudorandom value to sample from this distribution. We refer to (Piet et al., 2023) for more details.

## 3. Threat Model

We consider a provider capable of training LLMs and deploying them to many users via a black-box API, such as Google with Gemini or OpenAI with ChatGPT. The threat to the provider are untrustworthy users who misuse the provided LLM and generate harmful content without detection.

**Provider's Capabilities and Goals** *(Deployment)* The provider fully controls the LLM and its text generation process, including the ability to embed a watermark into generated text. *(Watermark Verification)* The provider must be able to verify their content watermark in each generated text. Their goal is to have a watermark that is (i) quality-preserving and (ii) robust, enabling detection of generated text at a given, low False Positive Rate (FPR) $\rho \in \mathbb{R}^+$.

**Attacker's Capabilities.** *(Access Restrictions)* We consider a (i) no-box attacker who cannot collect any watermarked texts during training and is (ii) offline, meaning that they cannot access the provider's VERIFY function. Our focus is on (iii) adaptive attackers, who know the provider's watermark algorithms (KEYGEN, EMBED, VERIFY) but do not know the secret inputs used for watermarking, such as random seeds or the provider's LLM. We also evaluate how adaptive attacks transfer in the non-adaptive setting against unseen watermarks. *(Surrogate Models)* A surrogate model is a model trained for the same task as the provider's model. For example, while GPT-4o's weights are not public, the attacker can access the parameters of smaller, publicly available models such as those from the `Llama2` (Touvron et al., 2023) model family. Our attacker can use such open-weight *surrogate* models for *paraphrasing* text. We assume the surrogate model's text quality is inferior to the provided model; otherwise, there would be no need to use the watermarked model. *(Compute)* Our attacker has limited computational resources and cannot train LLMs from scratch.

**Attacker's Goals.** The attacker wants to use the provided, watermarked LLM to generate text (i) without a watermark and (ii) with high quality. We measure text quality using many metrics, including a quality function $Q : \mathcal{V}^* \times \mathcal{V}^* \to \mathbb{R}$ between pairs of text when the attacker attempts to evade detection. We require that the provider correctly verifies their watermark with a given p-value threshold at most $\rho$. Lower thresholds make evasion more likely to succeed, i.e., detection becomes more challenging for the provider.

**Our motivation** is to evaluate the robustness of watermarking against constrained attackers that (i) have limited resources and (ii) lack any information about the watermarking key and samples. If successful attacks exist in this pessimistic no-box setting, the provider cannot hope to have a robust watermark against more capable attackers (e.g., with black-box access). We show that (i) such attacks exist, (ii) they are inexpensive, and (iii) they do not require access to watermarked samples. We believe the development of defenses should focus on the no-box setting first.

## 4. Related Work

We evaluate the robustness of *content* watermarking (Lukas & Kerschbaum, 2023) methods against no-box, offline attackers in the adaptive and non-adaptive settings (see Section 2). Other watermark evasion attacks, including those by Hu et al. (2024), Kassis & Hengartner (2024), and Lukas et al. (2024), focus on the image domain, whereas our work focuses on LLMs. Jovanović et al. (2024); Pang et al. (2024) propose black-box attacks against LLMs that require collecting many watermarked samples under the same key-message pair. We focus on no-box attacks. Jiang et al. (2023) propose online attacks with access to the provider's watermark verification, whereas we focus on a less capable *offline* attacker who cannot verify the presence of the provider's watermark. Current attacks are either non-adaptive, such as DIPPER (Krishna et al., 2023) or handcrafted against one watermark (Jovanović et al., 2024). We focus on optimizable, adaptive attacks and show that they remain effective in the non-adaptive setting.

Zhang et al. (2024) demonstrated the impossibility of robust watermarking against attackers with access to quality and perturbation oracles, showing that random walks with the perturbation oracle provably removes watermarks. Our approach differs in that it adaptively optimizes to find a single-step perturbation for evading watermark detection. We demonstrate the feasibility and efficiency of our attacks, achieving watermark evasion at low computational cost (USD $\leq 10\$$).

## 5. Conceptual Approach

Our goal is to adaptively fine-tune an open-weight paraphraser $\theta_P$ against known watermarking methods. The attacker lacks knowledge of the provider's watermarking key $\tau \leftarrow$ KEYGEN$(\theta, \gamma)$, which depends on (i) the unknown random seed $\gamma$ and (ii) the unknown parameters $\theta$ of the provider's LLM. Our attacker overcomes this uncertainty by choosing an open-weight surrogate model $\theta_S$ to generate so-called *surrogate* watermarking keys $\tau'$ and optimizes the expected evasion rate over many random seeds $\gamma \sim \mathbb{R}$.

## 5.1. Robustness as an Objective Function

Let $P_\theta : \mathcal{V}^* \to \mathcal{V}^*$ denote a randomized paraphrasing function[4], $H_\theta : \mathcal{V}^* \to \mathcal{V}^*$ is a function to generate text given a query $q \in \mathcal{T} \subseteq \mathcal{V}^*$ and $Q : \mathcal{V}^* \times \mathcal{V}^* \to \mathbb{R}$ measures the similarity between pairs of text. We formulate robustness using the objective function in Equation (2) that optimizes the parameters $\theta_P$ of a paraphrasing model.

$$\max_{\theta_P} \; \mathbb{E}_{\substack{\gamma \sim \mathcal{R} \\ m' \sim \mathcal{M} \\ q \sim \mathcal{T}}} \left[ \mathbb{E}_{\substack{\tau' \leftarrow \text{KEYGEN}(\theta_S, \gamma) \\ \theta_S^* \leftarrow \text{EMBED}(\theta_S, \tau', m') \\ x \leftarrow H(\theta_S^*, q) \\ x' \leftarrow P(\theta_P, x)}} \right. \tag{2}$$
$$\left. \text{VERIFY}\big(x', \tau', m'\big) \; + \; Q\big(x', x\big) \right].$$

Equation (2) finds optimal parameters for the paraphraser $\theta_P$ by sampling uniformly at random over (i) random seeds $\gamma \sim \mathbb{R}$, (ii) messages $m' \sim \mathcal{M}$ and (iii) queries $q \sim \mathcal{T}$. The second expectation is taken over a *surrogate watermarking key*, generated using knowledge of the KEYGEN algorithm, the surrogate model's parameters $\theta_S$ and a (previously sampled) random seed $\gamma$ as input. The surrogate model, key, and message are used to embed a watermark into the surrogate model $\theta_S^*$ (with knowledge of EMBED), which generates a watermarked sample $x$. The optimization process finds optimal parameters $\theta_P^*$ such that the paraphraser has a high probability of generating text $y \leftarrow P(\theta_P, x)$ that evades watermark detection and preserves text quality compared to $x$. Note that knowledge of the watermarking algorithms (KEYGEN, EMBED, VERIFY) is required to generate surrogate keys needed to optimize Equation (2).

Optimization presents multiple challenges. The attacker optimizes over different random seeds $\gamma$ and a surrogate model $\theta_S$ than those used by the provider, since our attacker does not know the provider's model parameters $\theta$ or random seeds. This lack of knowledge adds uncertainty for the attacker. The discrete nature of text and the inability to backpropagate through its generation process make maximizing the reward challenging (Shin et al., 2020). Furthermore, the reward function depends on VERIFY, which may not be differentiable. Deep reinforcement learning (RL) methods (Schulman et al., 2017; Rafailov et al., 2024) do not require differentiable reward functions. However, RL is known to be compute-intensive and unstable, making it unclear whether optimization can achieve a high reward using limited computational resources.

---

[4] We consider language models as paraphrasers, where randomness arises from sampling the next token.

**Algorithm 1** curates a preference dataset to optimize the adaptive attack's objective in Equation (2).

---
**Require:** Surrogate $\theta_S$, Paraphraser $\theta_P$, Queries $\mathcal{T}$, Messages $\mathcal{M}$, Paraphrase Repetition Rate $N$, False Positive Rate Threshold $\rho$, Quality Threshold $\delta$

1: $\mathcal{D} \leftarrow \emptyset$ // *The preference dataset*
2: // *Sample from known watermarking methods* $\mathcal{W}$
3: **for** (KEYGEN, EMBED, VERIFY) $\in \mathcal{W}$ **do**
4:     **for** each $q \in \mathcal{T}$ **do**
5:         $m \sim \mathcal{M}$
6:         $\tau' \leftarrow \text{KEYGEN}(\theta_S, \text{RND}())$
7:         $\theta_S^* \leftarrow \text{EMBED}(\theta_S, \tau', m)$
8:         $r \leftarrow S_{\theta_S^*}(q)$ // *Watermarked text under $\tau$'*
9:         // *If watermark can be detected*
10:         **if** VERIFY$(r, \tau', m) < \rho$ **then**
11:             // *Rejected (0) and Chosen (1) paraphrases*
12:             $R^0, R^1 \leftarrow \emptyset, \emptyset$
13:             **for** $i \in [N]$ **do**
14:                 $r' \leftarrow P_{\theta_P}(r)$ // *Paraphrase (randomized)*
15:                 $a \leftarrow \mathbf{1}[Q(r, r') \geq \delta]$
16:                 $b \leftarrow a \cdot \mathbf{1}[\text{VERIFY}(r', \tau', m) > \rho]$
17:                 $R^b \leftarrow R^b \cup \{r'\}$
18:             **end for**
19:             **for** $j \in [|R^1|]$ **do**
20:                 $r'_n \leftarrow (j \leq |R^0|) \; ? \; R^0_j : r$
21:                 $\mathcal{D} \leftarrow \mathcal{D} \cup \{(r, r'_n, R^1_j)\}$ // *Match pairwise*
22:             **end for**
23:         **end if**
24:     **end for**
25: **end for**
26: **return** $\mathcal{D}$ // *The preference dataset*

---

## 5.2. Preference Dataset Curation

We use reinforcement learning (RL) methods such as Direct Preference Optimization (DPO) (Rafailov et al., 2024) to optimize Equation (2). However, DPO requires collecting a preference dataset of *positive* and *negative* examples to fine-tune the paraphraser. A *negative* sample is one that retains the watermark, representing a failed attempt at watermark evasion. In contrast, positive samples do not retain a watermark and have a high text quality $Q(r, r'_p) > \delta$ for an attacker-chosen $\delta \in \mathbb{R}^+$. To bootstrap optimization, we require the ability to curate positive and negative examples, which we achieve by using a publicly available, open-weight paraphrasers such as Llama2-7b. We curate triplets $(r, r'_n, r'_p)$ via best-of-N rejection sampling. These triplets contain a watermarked sample $r$ and two paraphrased versions, $r'_n$ and $r'_p$, representing the negative and positive examples, respectively. Algorithm 1 implements the algorithm to curate our preference dataset.

Algorithm 1 randomly samples from a set of known watermarking methods $\mathcal{W}$ (line 3) and from the set of task-

specific queries $\mathcal{T}$ (line 4). It samples a message $m$ (line 5) and generates a surrogate watermarking key $\tau'$ to embed a watermark into the surrogate generator (lines 6-7). We generate text $r$ using the watermarked model $\theta_S^*$ (line 8) and verify whether it retains the watermark (line 9). The paraphrase model $\theta_P$ generates $N$ paraphrased versions of $r$ that we partition into positive and negative samples (lines 13-17). A sample $r_p'$ is positive ($b = 1$) if it does not retain the watermark and has high text quality ($\geq \delta$); otherwise, it is negative ($r_n'$, $b = 0$). For each positive sample, we select one corresponding negative sample and add the watermarked text and the negative and positive paraphrases to the preference dataset $\mathcal{D}$ (lines 19-21).

## 6. Evaluation

We report all runtimes on NVIDIA A100 GPUs accelerated using VLLM (Kwon et al., 2023) for inference and Deep-Speed (Microsoft, 2021) for training. Our implementation uses PyTorch and the Transformer Reinforcement Learning (TRL) library (von Werra et al., 2020). We use the open-source repository by Piet et al. (2023), which implements the four surveyed watermarking methods. We test robustness using hyper-parameters suggested by (Piet et al., 2023). Please refer to Section A.8 for details on hyperparameter selection and generalizability of our attacks against a range of hyperparameters. All LLMs used in our evaluations have been instruction-tuned. A detailed description of our attack setup, including prompting strategies and training hyperparameters, is available in Section A.9. Table 1 summarizes other surveyed evasion attacks.

### 6.1. Preference Dataset Collection

For a given watermarked sequence generated by the surrogate model, the attacker generates $N$ paraphrased versions using the non-optimized paraphraser and calculates the best-of-N evasion rate with the surrogate key (Algorithm 1, lines 9-12). Figure 2 shows the number of repetitions $c$ needed to achieve a given evasion rate across four watermarking methods using `Llama2-7b` as both the surrogate and paraphrasing model. Our attacker can choose the best-of-N paraphrases because they know the surrogate watermarking key to detect a watermark. The attacker cannot choose the best-of-N paraphrases against the provider's watermarked text, as they lack access to the provider's key.

Figure 2 shows the success rate of observing at least one positive sample after N paraphrases against methods designed for robustness (`Dist-Shift`, `Exp`) and undetectability (`Inverse`, `Binary`). The attacker requires limited computational resources to curate a large preference dataset against any of the four surveyed watermarks. For instance, to collect $|D| = 7\,000$ preference samples, each of $T = 512$ tokens, at a rate of $1\,800$ tokens/second, we expect this to

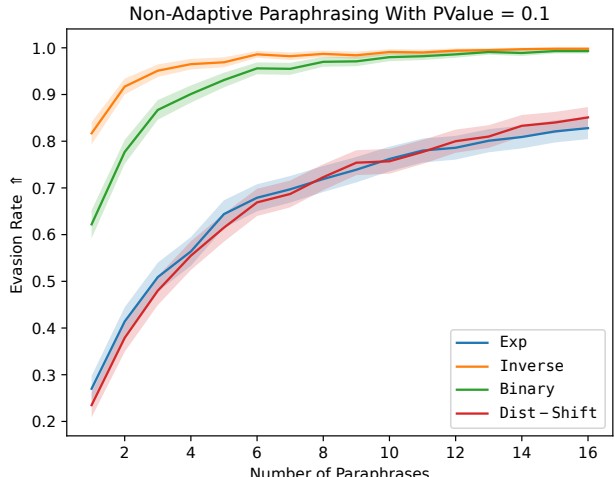

Figure 2: Algorithm 1 paraphrases text $N$ times in lines 13-17. This graph shows the expected evasion rate of the best sample (lines 15-17) for the number of paraphrases using a vanilla `Llama2-7b` as the paraphraser.

take approximately 1.5 GPU hours for `Dist-Shift`, but only 0.5 GPU hours for `Inverse`. In practice, including the overhead of evaluating quality and detecting watermarks, we require less than 5 GPU hours to curate $7\,000$ samples for `Dist-Shift`. At current AWS rates, an attacker who uses our attacks faces only negligible costs of less than \$10 USD to curate a preference dataset containing $7\,000$ samples and fine-tune the paraphraser. Further details on the curation of the prompt sets used for training and evaluation are provided in Section A.2.

### 6.2. Ablation Studies

In our experiments, we ablate over the following settings.

(1) **Adaptivity**: (*Adaptive*) The same watermarking method is used for training and testing. (*Non-adaptive*) The attack is tested against unseen watermarking methods., (2) **Target Models**: We evaluate 2 models used by the provider: `Llama2-13b`, `Llama3.1-70b`., (3) **Attacker's Models**: Our attacker matches surrogate and paraphrasing models. We consider `Llama2` (Touvron et al., 2023) and `Qwen2.5` (Qwen, 2024) from 0.5b to 7b parameters., (4) **Watermarking Methods**: `Exp` (Aaronson & Kirchner, 2023), `Dist-Shift` (Kirchenbauer et al., 2023b), `Inverse` (Kuditipudi et al., 2024), `Binary` (Christ et al., 2023)., (5) **Hyper-Parameters**: We ablate over multiple hyper-parameters that a provider can choose (see Section A.8)., and (6) **False Positive Rates (FPR)**: Section A.10 ablates over $\rho \in \{0.01, 0.025, 0.05, 0.075, 0.1\}$ when the provider can tolerate higher FPR thresholds for detection.

A watermark has been *retained* if the null hypothesis that

| Attack Name | Description |
|---|---|
| DIPPER (Krishna et al., 2023) | Train an 11b Sequence-to-Sequence model for paraphrasing. |
| Translate (Piet et al., 2023) | Translate to another language and back (e.g., French, Russian). |
| Swap (Piet et al., 2023) | Randomly remove, add or swap words. |
| Synonym (Piet et al., 2023) | Replace words with a synonym using WordNet (Miller, 1995). |
| HELM (Bommasani et al., 2023) | Randomly add typos, lowercase or contractions. |
| Llama, Qwen2.5, GPT3.5 | Paraphrase text using a publicly accessible LLM. |
| Ours-`Llama2-7b-Exp` | Paraphrase with a `Llama2-7b` model tuned adaptively against `Exp`. |

Table 1: (Top) The non-adaptive baseline attacks we consider in our study against (Bottom) our adaptively fine-tuned attacks. We refer to (Piet et al., 2023) for details on the baseline attacks and Section A.9 for our adaptive attack.

the watermark is not present in the content can be rejected with a given p-value specified by the provider. The *evasion rate* is calculated as the fraction of watermarked text that does not retain the watermark after applying the paraphrasing attack. Due to the lack of a gold-standard metric to assess text quality, we measure quality with multiple metrics: LLM-Judge, LLM-CoT, and LLM-Compare from Piet et al. (2023), Mauve (Pillutla et al., 2021), and Perplexity (PPL) with `Llama3-8B-Instruct`. To enhance clarity, we only report the LLM-Judge metric in the main paper following Piet et al. (2023). Full descriptions of all quality metrics are provided in Section A.1. Unless otherwise specified, we use a p-value threshold of $\rho = 0.01$.

### 6.3. Experimental Results

**Adaptivity**. Figure 3 shows the evasion rate and text quality of our methods trained in the adaptive and non-adaptive settings when the provider uses `Llama2-13b` and the attacker uses `Llama2-7b`. We find that all adaptive attacks have an evasion rate of at least 96.6%, while the non-adaptive attacks have an evasion rate of at least 94.3%. We achieve the highest overall evasion rate when training against the `Exp` watermark, which achieves an evasion rate of at least 97.0%. We train one attack, denoted `All`, against all four surveyed watermarking methods and test it against each watermark separately. Interestingly, `All` performs slightly worse compared to training only on `Exp`, exhibiting an evasion rate of at least 96.3% and a lower paraphrased text quality of at least 0.893 (versus 0.901 when training only on `Exp`). In summary, Figure 3 shows that adaptive attacks trained against one watermark remain highly effective when tested against unseen watermarks in the non-adaptive setting.

**Model Sizes.** Figure 4 shows the Pareto front against the `Exp` watermark with a `Llama3.1-70b` target model. Our attacker uses paraphraser models with at most 7b parameters, which is less than the 11b DIPPER model (Krishna et al., 2023) currently used to test robustness.

We observe that (1) **Non-adaptive baseline** attacks such as Contraction, Swapping and Synonym replacements are ineffective and have a low evasion rate of less than 20%.,

(2) **Non-adaptive model-based** paraphrasing attacks such as using vanilla `Llama2-7b` or `ChatGPT3.5` models have a substantially higher evasion rate of 61.8% up to 86.1% respectively. Tuning `Llama2-7b` using our approach in the non-adaptive setting improves the evasion rate substantially to 90.9% (when trained on `Binary`) and up to 97.6% (when trained on `Inverse`). These non-adaptive, optimized attacks have a paraphrased text quality of 0.853 and 0.845, slightly improving over `ChatGPT3.5`, rated only 0.837., and (3) **In the adaptive setting**, our fine-tuned `Qwen2.5-7b` achieves an evasion rate of 97.3% at the highest text quality of 0.846 compared to `Llama2-7b-Inverse`.

By ablating over `Qwen2.5` between 0.5b and 7b parameters, we find that attackers can strictly improve paraphrased text quality at similar evasion rates by using more capable paraphrases with more parameters. Figure 16 in the Appendix shows results against a `Llama2-13b` target model, which are consistent with those against `Llama3.1-70b`. Against smaller target models, attackers can achieve higher evasion rates and text quality ratings.

**Text Quality**. Table 2 shows (i) a watermarked text sample generated using `Llama2-13b` with `Dist-Shift`, (ii) paraphrased text using a non-optimized `Llama2-7b` model, and (iii) paraphrased text obtained with an adaptively tuned `Llama2-7b` model using our attack. We observe that all paraphrased texts preserve quality, but our attack achieves the lowest green-to-red token ratio (i.e., maximizes the evasion rate). Table 3 in the Appendix shows a quantitative analysis of the median quality of generated text for a vanilla `Llama2-7b` model compared to our best adaptive and non-adaptive attacks. It shows that text quality is preserved across five text quality metrics when using our attacks. We only show one paragraph of generated text that we truncated due to space restrictions and Tables 5 and 6 in the Appendix show non-truncated samples. Table 6 shows a rare, cherrypicked example where our attack fails at evading watermark detection after paraphrasing.

**Adaptive vs Non-adaptive.** Figure 5 shows two results to compare the non-optimized `Llama2-7b` with our adap-

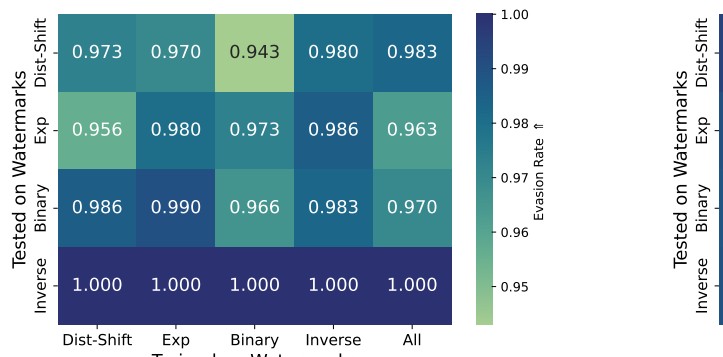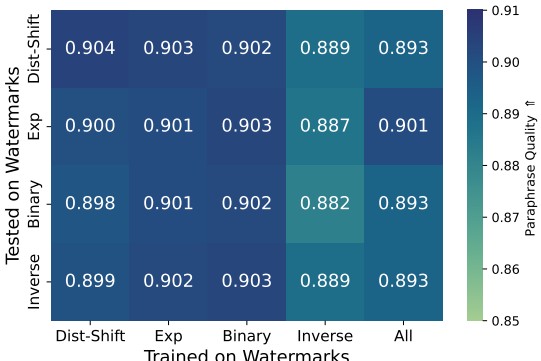

Figure 3: The evasion rates (Left) and text quality measured with LLM-Judge (Right). The attacker uses a matching `Llama2-7b` surrogate and paraphraser model versus the provider's `Llama2-13b`. Results for adaptive attacks are on the diagonal. For example, we obtain the bottom left value by training on `Dist-Shift` and testing on `Inverse`.

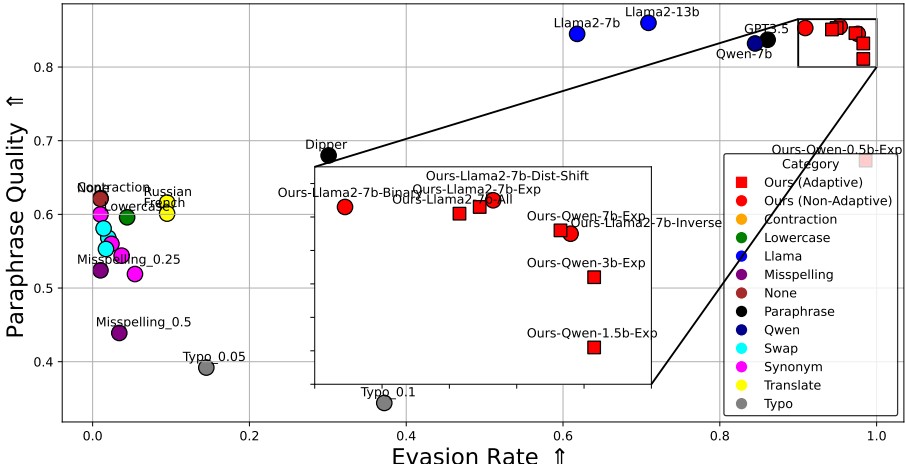

Figure 4: Adaptive attacks are Pareto-optimal. We show the evasion rate versus text quality trade-off against the `Exp` (Aaronson & Kirchner, 2023) watermark, corresponding to $(\epsilon, \delta)$-robustness from Eq. 1. The provider uses a `Llama3.1-70b` model, whereas our attacker's models are up to $46\times$ smaller. Non-adaptive attacks are marked by circles ($\bigcirc$), adaptive attacks by squares ($\square$). Notation "`Ours-Qwen-3b-Exp`" means that we evaluate our attack using a `Qwen2.5-3b` model that was adaptively optimized against the `Exp` watermark.

tively tuned `Llama2-7b` model. The result on the left plots the cumulative density of p-values. Our method strictly improves over the non-optimized model as it generates paraphrased text with higher mean p-values for watermark detection. The result on the right plots the expected p-value against the token length. The watermarked text has a median p-value of less than $0.01$ after approximately 170 tokens, whereas the non-optimized `Llama2-7b` model has an expected p-value of $0.10$ at around 500 tokens compared to an expected p-value of $0.4$ for our adaptively tuned model.

**Additional Testing.** We present more results to compare adaptive versus non-adaptive attacks in Section A.4, including tests against other recently released watermarking methods. These results are consistent with our findings in the main part of the paper that adaptive attacks are Pareto

optimal and outperform much larger, closed-source systems such as GPT-4o at evading watermark detection. We kindly refer the reader to Section A.4 for more baseline tests, Section A.5 for further statistical insights, and Section A.6 for an analysis of the impact of paraphrasing on the top-50 token distribution.

## 7. Discussion

**Effectiveness of Adaptive Attacks.** Our work demonstrates that content watermarking methods for LLMs are vulnerable to adaptively optimized attacks. Attackers can adaptively fine-tune relatively small open-weight models, such as `Llama2-7b` (Touvron et al., 2023), in less than seven GPU hours to evade watermark detection from larger and

| Llama2-13b-Dist-Shift | Llama2-7b | Ours-Llama2-7b-Dist-Shift |
|---|---|---|
| Cervantes' "Don Quixote" is a ground-breaking masterpiece of literature that has enthralled readers for over 400 years. The novel tells the story of Alonso Quixano, an old man who becomes so consumed by the chivalrous tales of his day that he renames himself Don Quixote and embarks upon an adventurous journey filled with humor, satire, and timeless themes about the human condition.↩↩ The story follows Quixote and his trustworthy sidekick Sancho Panza, a simple and pragmatic squire who provides moments of humor and contrast to the lofty ideals of the delusional Quixano/Don Quixote. Their travels in rural Spain lead them to encounter all sorts of people they mistake for knights, ladies in distress, bandits, and giants who are in fact ordinary villagers going about their everyday lives. | "Don Quixote" is a groundbreaking master-piece of literature that has captivated read-ers for over 400 years. The novel tells the story of Alonso Quixano, an elderly man whose fixation on chivalrous tales leads him to change his name to Don Quixote and embark on a thrilling adventure replete with humor, satire, and timeless themes concerning the human condition.↩↩ The narrative follows Don Quixote and his loyal squire Sancho Panse, a practical and good-natured individual who provides mo-ments of levity and contrast to the lofty ide-als of the delusional Quixano/Don Quixote. Their travels across rural Spain result in encounters with various people who they misidentify as knights, distressed ladies, bandits, and ordinary villagers going about their daily lives. | Spain has been blessed with a literary treasure that has captivated readers for over four centuries in the form of Miguel de Cervantes' immortal masterpiece, "Don Quixote". This timeless tome weaves a hilarious and poignant tale of a man con-sumed by the chivalrous stories of his youth, who renames himself Don Quixote and sets forth on a journey filled with mo-ments of satire and a piercing examina-tion of the human condition.↪↪As Don Quixote and his trusty squire Sancho Panza traverse rural Spain, they encounter vari-ous unsuspecting villagers, mistaking them for knights, maidens in distress, bandits, and even giants. |
| 163 green and 36 red | 125 green and 69 red | 81 green and 78 red |

Table 2: (Left) Watermarked text from the provider's `Llama2-13b` model, (Center) a paraphrased version from a vanilla `Llama2-7b` model, and (Right) paraphrased text using our adaptively tuned `Llama2-7b` model. Green/red indicates whether a token is watermarked. A lower green-to-red token ratio implies a higher evasion rate. Due to space constraints, we only show truncated texts. Tables 5 and 6 in the Appendix show entire samples with up to 512 tokens.

more capable models, such as `Llama3.1-70b` (Dubey et al., 2024). Our attacks remain effective even in the non-adaptive setting when testing with unseen watermarking methods. Our findings challenge the robustness claims of existing watermarking methods, and we propose improved methods to test robustness using adaptive attacks.

**Analysis.** Studying *why* adaptive attacks work is challenging due to the non-interpretability of the optimization process. The ability to maximize Equation (2) implies the ability to evade detection since Equation (2) encodes robustness for any watermarking method. The effectiveness of non-adaptive attacks could be explained by the fact that all surveyed watermarks are similar in that they operate on the token level. Hence, an effective attack against one watermark likely generalizes to other unseen watermarks. Adaptive attacks further improve effectiveness as there are at least three learnable signals for paraphrasing watermarked text: (1) Avoid repeating token sequences, as they likely contain the watermark; (2) find text replacements with low impact on text quality to maximize the evasion rate (e.g., uncommon words or sentence structures); and (3) calibrate the minimum token edit distance and lexical diversity that, on average (over the randomness of the key generation process), evades detection. We refer to Section A.7 for a more detailed analysis of our approach's effectiveness.

**Attack Runtime.** Our attacks involve two steps: Dataset Curation and Model Optimization. Curating 7 000 samples requires less than 5 GPU hours, and model optimization requires only approximately 2 GPU hours for a `Llama2-7b` model at 16-bit precision. These attacks can be executed with limited computational resources and cost less than $10 USD with current on-demand GPU pricing.

**Restricted Attackers.** Zhang et al. (2024) show that *strong* watermarking, which resists any attack, is provably impossible under certain conditions. Our work instead focuses on robustness against restricted attackers with limited capabilities, such as limited compute resources, and we study whether robustness can be achieved in this setting. We show that current watermarks do not achieve robustness, and that even restricted attackers can evade detection at low costs.

**Online Attacks.** Our work focuses on *offline* attacks that do not require any access to the provider's watermark detection functionality. Offline attacks evaluate the robustness of a watermark without any information about the secret key generated by the provider. An *online* attacker can learn information about the provider's secret key through accessing `Verify`, which reduces the attack's uncertainty and could substantially improve the attack's effectiveness further.

**Limitations.** Our study also did not focus on evaluating

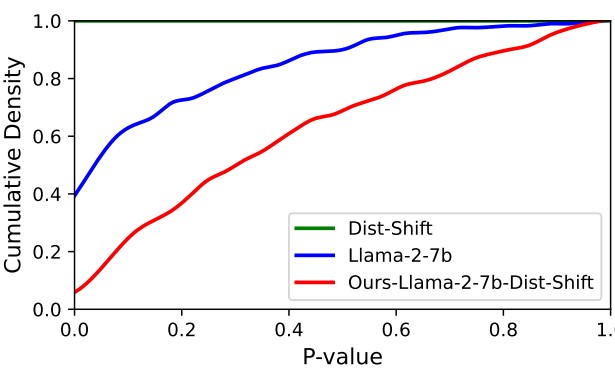 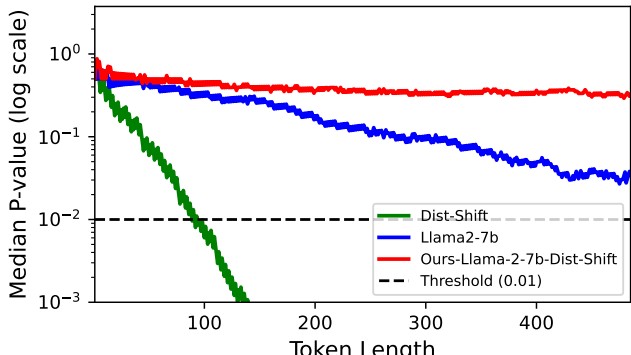

Figure 5: (Left) The cumulative density of p-values on the `Dist-Shift` watermark (green), a vanilla `Llama2-7b` paraphraser (blue) and our adaptive `Llama2-7b` paraphraser (red). (Right) The median p-value relative to the text token length with a threshold of $\rho = 0.01$ (dashed line).

adaptive defences that could be designed against our adaptive attacks. Adaptive defences have not yet been explored, and we advocate studying their effectiveness. We believe our optimizable, adaptive attacks will enhance the robustness of future watermarking methods by including them in their design process, for instance, by using adversarial training. We focused exclusively on text generation tasks and did not explore other domains, such as source code generation or question-answering systems, where different text quality metrics may be used to evaluate an attack's success. We did not consider the interplay between watermarking and other defenses, such as safety alignment or content filtering, which could collectively control misuse.

We acknowledge that LLM-as-a-Judge is an imperfect and noisy metric that may not align with human judgment. In the main part of our paper, we use `Llama3-8B`-as-a-Judge, since this metric is easily reproducible. Section A.4 shows results using GPT-4o-mini-as-a-Judge, which are consistent with our findings. More work is needed to study the metric's alignment with human judgment.

## 8. Conclusion

Our work demonstrates that current LLM watermarking methods are not robust against adaptively optimized attacks. Even resource-constrained attackers can reliably ($\geq 96.7\%$) evade detection with computational resource costs of $\leq\$10$ USD. They can achieve this with open-weight models that are $46\times$ smaller than the provider's model. Even in the non-adaptive settings, our adaptively tuned attacks outperform all other surveyed attacks, including paraphrasing with substantially larger models such as OpenAI's GPT4o. Our findings challenge the security claims of existing watermarking methods and show that they do not hold even against resource-constrained attackers. We suggest that future defenses must consider adaptive attackers to test robustness.

## Impact Statement

This work investigates the robustness of watermarking methods for large language models (LLMs), which has implications for content authentication and the responsible deployment of AI systems. Our findings demonstrate that attackers with limited computational resources can undermine the robustness of current watermarking methods by using adaptively optimized attacks. This vulnerability could have societal implications as major AI providers increasingly adopt watermarking to promote responsible AI use and control misuse, including the proliferation of LLM-generated misinformation and online spam.

By publicly releasing our methods, findings, and source code, we hope to encourage the development of more robust watermarking methods that can better withstand adaptive attacks (e.g., by increasing the computational complexity of such attacks or making them less effective). We acknowledge that our research could potentially be misused to evade existing deployments of watermarks. However, these deployments are still in experimental phases. We believe the benefit of releasing our work outweighs the potential harm and hope that our work inspires the development of more robust content authentication methods.

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

# A. Appendix

## A.1. Quality Metrics

Ideally, to evaluate the quality of an LLM-generated text, one would need a set of human evaluators, each giving their own score according to a certain rubric, and then have all the scores aggregated. However, this is impractical for both the attacker and the defender. Therefore, we employ multiple surrogate metrics from the literature: LLM-Judge, LLM-CoT, and LLM-Compare from (Piet et al., 2023), Mauve (Pillutla et al., 2021), and Perplexity (PPL) with `Llama3-8B-Instruct`. Note that all of these are implemented in the `MarkMyWords (MMW)` (Piet et al., 2023) benchmark utilized for our experiments. All the metrics evaluate a response (whether watermarked or a perturbed sample) against a baseline (either the original prompt, a non-watermarked sample, or the model's logit distribution). Below is a description of each metric, along with an indication of whether higher or lower values are better for that metric.

**LLM-Judge:** LLM-Judge directly prompts an instruction-tuned large language model to evaluate the quality of a certain response with respect to its original prompt. The response is decoded greedily (temperature =0) to ensure deterministic results. Criteria evaluated include: accuracy, level of detail, and typographical, grammatical, and lexical correctness. A higher score is better. For this, we use `Llama3-8B-Instruct` to evaluate, using the following prompt:

> **LLM-Judge Prompt**
>
> ```
> You are given a prompt and a response, and you provide a grade out of 100
> measuring the quality of the response, in terms of accuracy, level of details, and
> typographical, grammatical and lexical correctness.  Remove points as soon as one of
> the criteria is missed.  Prompt:{}, Response:{}
> ```

**LLM-CoT (Chain-of-Thought):** LLM-CoT evaluates the quality of the watermarked/attacked responses using CoT-based reasoning (Wei et al., 2023). A higher score is better. For this, we also use `Llama3-8B-Instruct` to evaluate, using the following prompt:

> **LLM-CoT Prompt**
>
> ```
> Please act as an impartial judge and evaluate the quality of the response provided
> by an AI assistant to the user question displayed below.  Your evaluation hinges
> on the following factors:  accuracy, level of details, typographical correctness,
> grammatical correctness and lexical correctness.  Be as objective as possible, and
> account for every one of these criteria in our evaluation.  Begin your evaluation
> by providing a short explanation.  After providing your explanation, please rate the
> response on a scale of 1 to 100 by strictly following this format:  "[[rating]]",
> for example:  "Rating:  [[51]]".  Your rating must come AFTER your explanation, not
> before.  Prompt:{}, Response:{}
> ```

**LLM-Comparator:** The LLM-Comparator is used to compare non-watermarked baseline response and the watermarked or attacked response. 0 indicates that the non-watermarked response is better, 0.5 indicates a tie, and 1 shows that the watermarked or attacked response is better. For this, we also use `Llama3-8B-Instruct` to evaluate, using the following prompt:

---

**LLM-Comparator Prompt**

```
 Please act as an impartial judge and evaluate the quality of the responses
provided by two AI assistants to the user question displayed below.  You should
choose the assistant that follows the user's instructions and answers the user's
question better.  Your evaluation should consider factors such as the helpfulness,
relevance, accuracy, depth, creativity, and level of detail of their responses.
Also account for typographical correctness, grammatical correctness and lexical
correctness.  Begin your evaluation by comparing the two responses and provide a
short explanation.  Avoid any position biases and ensure that the order in which
the responses were presented does not influence your decision.  Do not allow the
length of the responses to influence your evaluation.  Do not favor certain names
of the assistants.  Be as objective as possible.  After providing your explanation,
you must output your final verdict by strictly following this format:  * "[[A]]" if
assistant A is better, * "[[B]]" if assistant B is better, and * "[[C]]" for a tie.
For example, "Verdict:  [[C]]".  Prompt:  {}, [[Start of Assistant A]] {} [[End of
Assistant A's Answer]], [[Start of Assistant B]] {} [[End of Assistant B's Answer]]
```

**MAUVE:** MAUVE measures the similarity between two text distributions. In our case, the two distributions are the non-watermarked baseline response and the watermarked/paraphrased response. Higher MAUVE scores indicate that both texts match their content, quality and diversity. MAUVE is computed with the Kullback-Leibler (KL) divergences between two distributions in a lower-dimensional latent space. It correlates with human evaluations over baseline metrics for open-ended text generation (Pillutla et al., 2021). We use the `gpt2-large` model to compute the MAUVE score in our implementation.

**Perplexity (PPL):** Perplexity is a common language modelling metric that quantifies how well a model predicts a text sample. It is calculated based on the probability that the model assigns to a sequence of words. Lower perplexity values indicate that the model is more confident and accurate in its predictions, making lower scores better for this metric.

Table 3 shows the median text quality metrics to compare the vanilla `Llama2-7b` paraphraser to our best adaptive and non-adaptive attacks against the `Llama2-13B` and `Llama3.1-70B` target models. The table shows that our attacks have similar quality to the vanilla `Llama2-7b` paraphraser across the board. Our attacks have a higher MAUVE score, indicating that our text is closer to the non-watermarked text than the vanilla `Llama2-7b` paraphraser. The higher perplexity is not a concern, as it merely indicates that the large language model does not expect the text.

| Target: `Llama2-13B` | LLM-Judge ⇑ | LLM-CoT⇑ | LLM-Compare⇑ | Mauve⇑ | PPL⇓ |
|---|---|---|---|---|---|
| `Llama2-7b` | 0.92 | 0.85 | 0.00 | 0.17 | 4.74 |
| Ours-Best-Adaptive | 0.92 | 0.85 | 1.00 | 0.42 | 6.69 |
| Ours-Best-Non-Adaptive | 0.92 | 0.85 | 0.50 | 0.37 | 6.32 |
| Target: `Llama3.1-70B` | | | | | |
| `Llama2-7b` | 0.95 | 0.72 | 0.00 | 0.22 | 4.84 |
| Ours-Best-Adaptive | 0.95 | 0.72 | 0.50 | 0.55 | 6.10 |
| Ours-Best-Non-Adaptive | 0.95 | 0.72 | 0.50 | 0.31 | 6.15 |

Table 3: Various median text quality metrics to compare the vanilla `Llama2-7b` paraphraser to our best adaptive and non-adaptive attacks. We limit all attacks to at most 7b parameter models.

## A.2. Prompt-set Curation

The **evaluation set** consists of 296 prompts from Piet et al. (2023), covering book reports, storytelling, and fake news. The **training set** comprises a synthetic dataset of 1 000 prompts, covering diverse topics including reviews, historical summaries, biographies, environmental issues, science, mathematics, news, recipes, travel, social media, arts, social sciences, music, engineering, coding, sports, politics and health. To create this dataset, we repeatedly prompt a large language model (ChatGPT-4) to generate various topic titles. These titles were then systematically combined and formatted into prompts.

The synthetic training dataset is non-overlapping with the evaluation set. Nonetheless, in realistic scenarios, it is plausible that an attacker might train and evaluate their paraphraser using the same dataset. Given the low cost of our attack (USD $\leq 10\$$), attackers can easily train their own paraphrasers.

### A.3. Preference-data Curation

For every prompt in the training set, we generate water-marked output using each watermark; then, we use that output as input to our paraphrasers. Each paraphraser generates 16 paraphrases for each input. We then filter these paraphrases as per Algorithm 1 to create the training preference pairs. Larger models have higher quality output and so have a higher yield of successful paraphrases. We use the same number of paraphrases for each model, even though they may generate different yields.

Figure 6 shows the expected evasion rate versus the number of paraphrases ablated over varying model sizes of `Qwen2.5` (Qwen, 2024) against the `Exp` watermark. We find that the expected evasion rate increases with the number of paraphrases, but the rate of increase diminishes as the number of paraphrases increases. We find that the expected evasion rate does not improve significantly close to 16 paraphrases and that bigger models tend to have higher evasion rates for the same number of paraphrases. An exception to this is the 1.5b model, which surprisingly performs very well (better than the 3b) for the same number of paraphrases. This, however, could be due to different pretraining parameters of the base model or other factors.

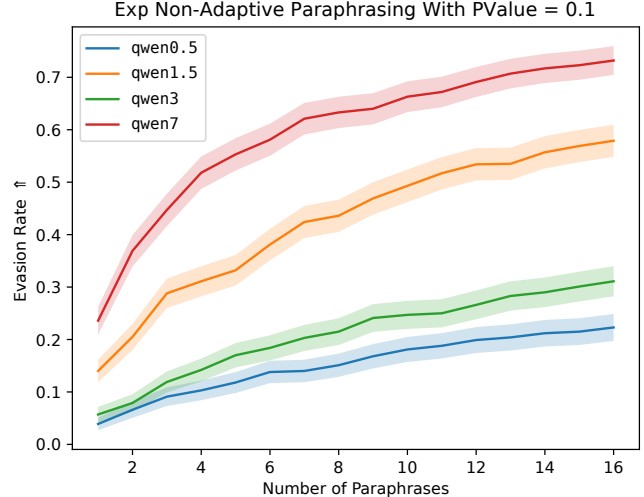

Figure 6: The expected evasion rate versus the repetition rate ablated over varying model sizes of `Qwen2.5` (Qwen, 2024) against the `Exp` watermark (Algorithm 1, lines 9-12). Shaded areas denote 95% confidence intervals.

### A.4. Baseline Testing against other Watermarks

We include more robustness tests against recently released watermarks such as SynthID (Dathathri et al., 2024), Unigram (Zhao et al., 2024) and SIR (Liu et al., 2024). We refer to the author's papers for detailed descriptions of these watermarks. For this evaluation, we implemented our attack in the MarkLLM framework (Pan et al., 2024), used our `Qwen2.5-3b` paraphraser trained against the EXP watermark from the main part of the paper, and adaptively tuned a new `Qwen2.5-3b` paraphraser against the Unigram watermark. For the purpose of quick evaluation, we limit the token length to 256 tokens, noting that, as shown in Figure 5, the results are similar for longer texts. GPT-4o is part of the Pareto front only against SIR and KGW due to its high text quality and low evasion rates of less than 90%. It is not part of the Pareto front against SynthID, EXP and Unigram, where only our attacks are part of the Pareto front. While it may be possible to use better prompts for GPT-4o to achieve a higher text quality or evasion rate, there are other limitations when using closed systems to evade detection.

1. Their usage can be expensive as the user is typically charged per token.

2. The system could embed its own watermark into paraphrased text.

3. There could be guardrails such as safety alignments which prevent these systems from arbitrarily paraphrasing text.

In contrast, our method allows working with relatively small open-weight models that adversaries can fully control.

### A.5. Additional Statistics

We provide additional statistical insights complementing the robustness tests described in Section A.4. For brevity and clarity, we illustrate the statistics primarily with the Unigram watermark (Zhao et al., 2024), noting similar results across other watermark methods.

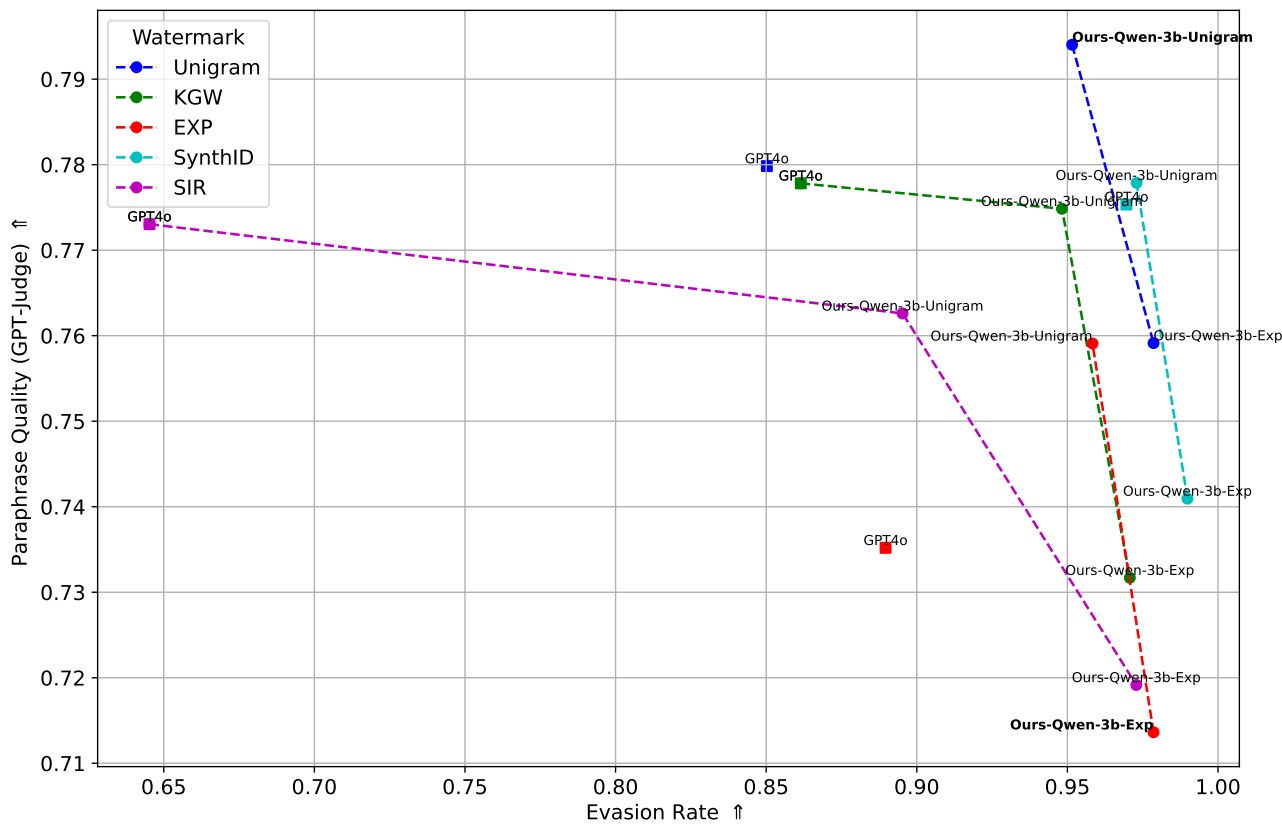

Figure 7: Additional results using `Qwen2.5-3b` against KGW and EXP, which we study in the main part of the paper, and more recently released watermarks such as SynthID (Dathathri et al., 2024), Unigram (Zhao et al., 2024) and SIR (Liu et al., 2024). Dashed lines denote the Pareto front, and we highlighted adaptively trained attacks in bold. We used GPT-4o's version from November 23rd, 2024. The y-axis uses GPT-4o-mini as a judge, and the x-axis shows the evasion rate.

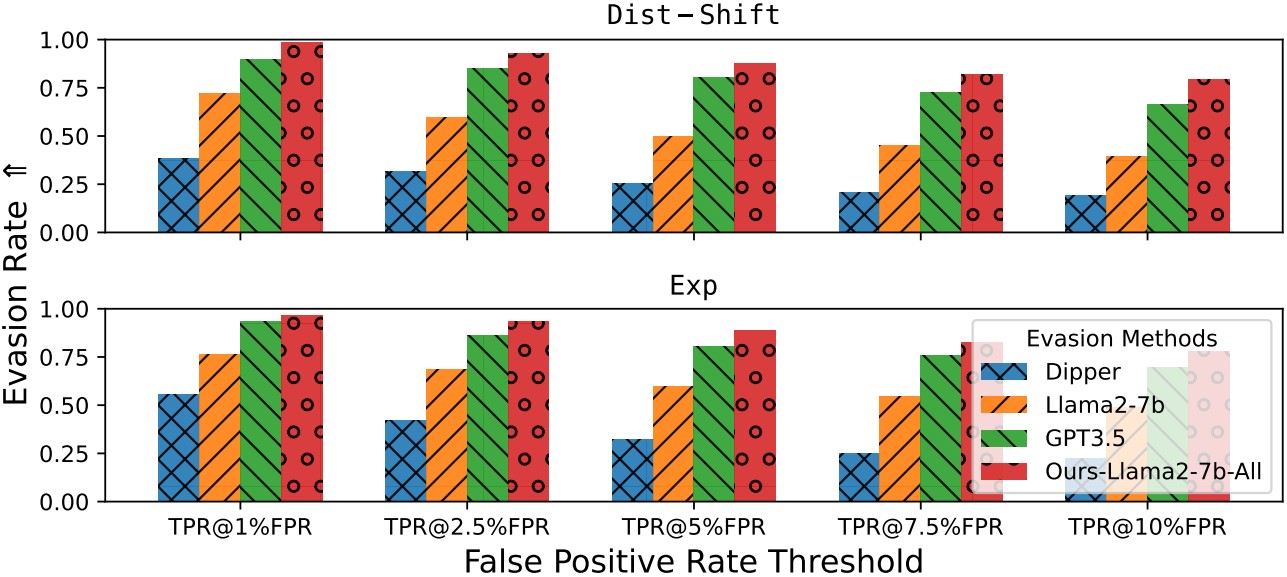

Figure 8: The evasion rates against a watermarked `Llama2-13b` model. We compare non-adaptive attacks, including ChatGPT3.5, versus our adaptively fine-tuned `Llama2-7b` paraphraser model.

**Token Length Analysis**    Figure 9 shows the distribution of token lengths for watermarked texts and different perturbations. Our tuned paraphrasers (`Qwen2.5-3b-Unigram` and `Qwen2.5-3b-EXP`) produce slightly shorter paraphrases compared to the base `Qwen2.5-3b` model and the watermarked responses themselves. This reduction in length likely arises from the optimization objective, which does not explicitly penalize brevity. Such behavior could be adjusted by modifying the objective function when selecting positive and negative samples. Non-optimized methods exhibit varied token lengths: GPT-3.5 generates even shorter responses, GPT-4o produces relatively longer texts, while word substitution (Word-S) and deletion (Word-D) methods behave as expected, respectively increasing or decreasing token counts.

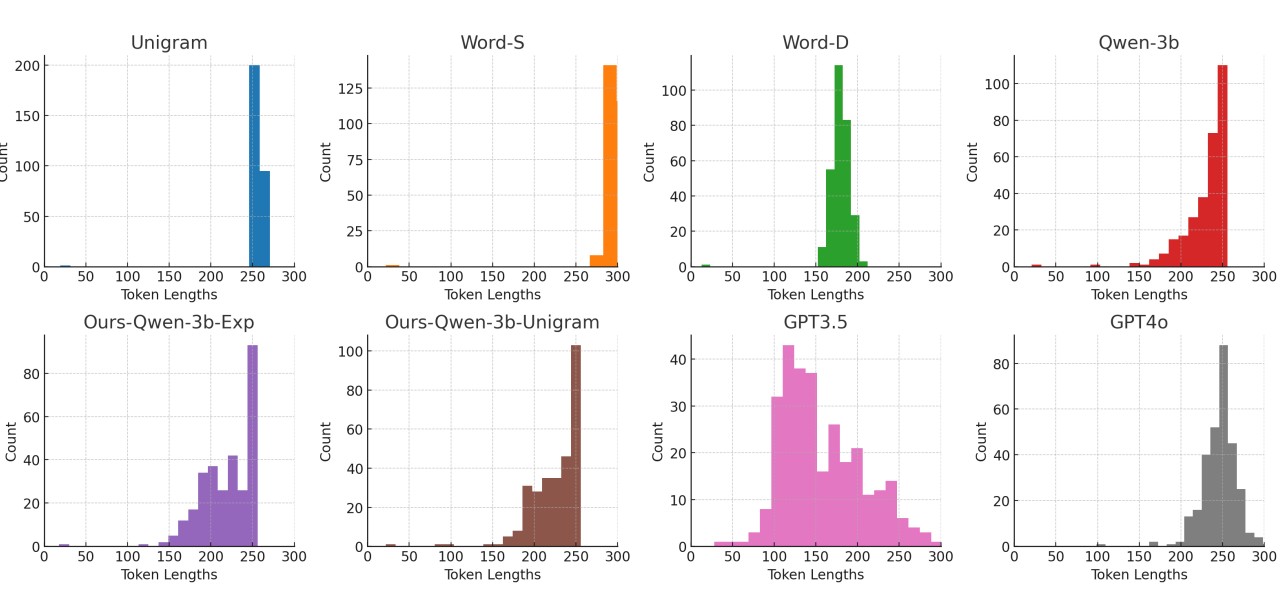

Figure 9: The distribution of the number of tokens in the watermarked text and the paraphrased texts. The x-axis shows the number of tokens, and the y-axis shows the number of samples.

**GPT-Judge Quality Scores**    The GPT-Judge scores are evaluated using GPT-4o-mini. Figure 10 indicates text quality across methods. Optimized paraphrasers (`Qwen2.5-3b-Unigram` and `Qwen2.5-3b-EXP`) exhibit similar high-quality scores to those of the unattacked Unigram watermark, base `Qwen2.5-3b`, GPT-3.5, and GPT-4o methods. In contrast, simple perturbations like Word-S and Word-D achieve significantly lower quality scores.

**Watermark Scores**    Figure 11 illustrates watermark detection scores as measured by the MarkLLM framework (Pan et al., 2024). The unattacked watermarked texts have notably high scores (centered around 5). Simple perturbations, such as word deletions, have no impact, while word substitutions moderately reduce scores. Non-tuned paraphrasing methods (`Qwen2.5-3b`, GPT-3.5, GPT-4o) substantially lower watermark scores (centered around 1). Adaptively fine-tuned paraphrasers (`Qwen2.5-3b-EXP` and `Qwen2.5-3b-Unigram`) achieve the lowest scores, typically centered around -1, highlighting their effectiveness in evading detection.

### A.6. Token Distribution

**Text Quality.**    Figures 12 to 14 show the top-50 token distribution that appear in the watermarked text. We compare it with the token frequency in the paraphrased text using as paraphrasers (i) GPT-4o, (ii) a baseline `Qwen2.5-3b` model and (iii) our adaptively tuned `Qwen2.5-3b` model against the Unigram watermark (Zhao et al., 2024). We observe that all paraphrasers have a similar token distribution and that across all three paraphrasers, on average, the top 50 tokens appear less frequently than in the original, watermarked text. The largest difference we observe between the baseline `Qwen2.5-3b` and our adaptively tuned model are the frequencies of the tokens 'The' and ' ' (space between words), which our model uses less frequently. Compared to GPT-4o, the baseline `Qwen2.5-3b` model uses some tokens, such as ' As', less frequently, while other tokens, such as ' but', appear more frequently.

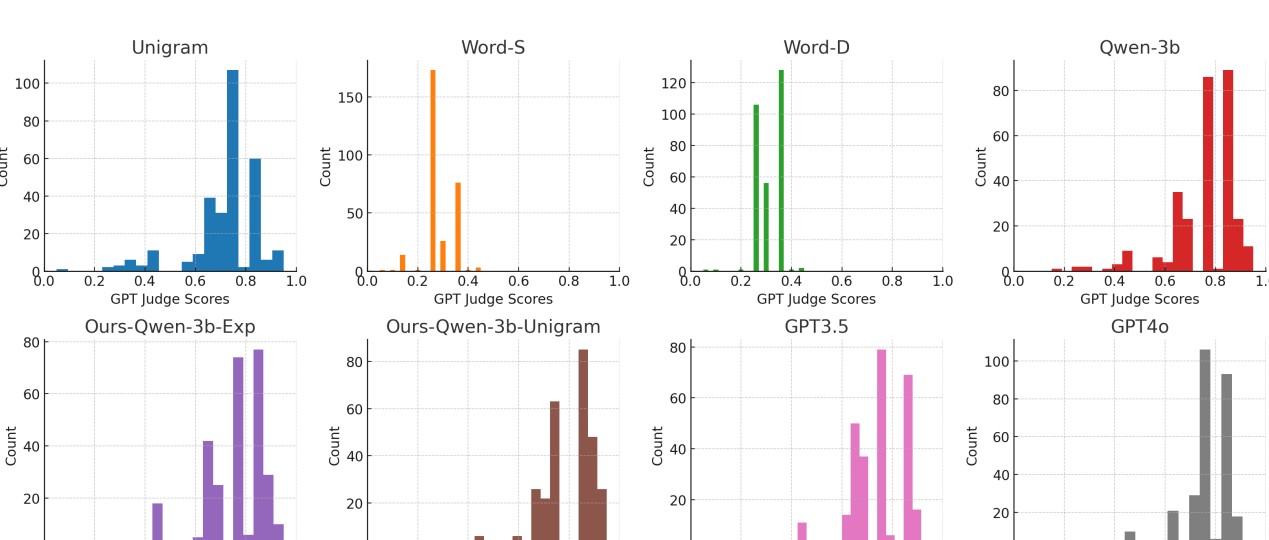

Figure 10: The distribution of the GPT-Judge scores for the watermarked text and the paraphrased texts. The x-axis shows the GPT-Judge score, and the y-axis shows the number of samples.

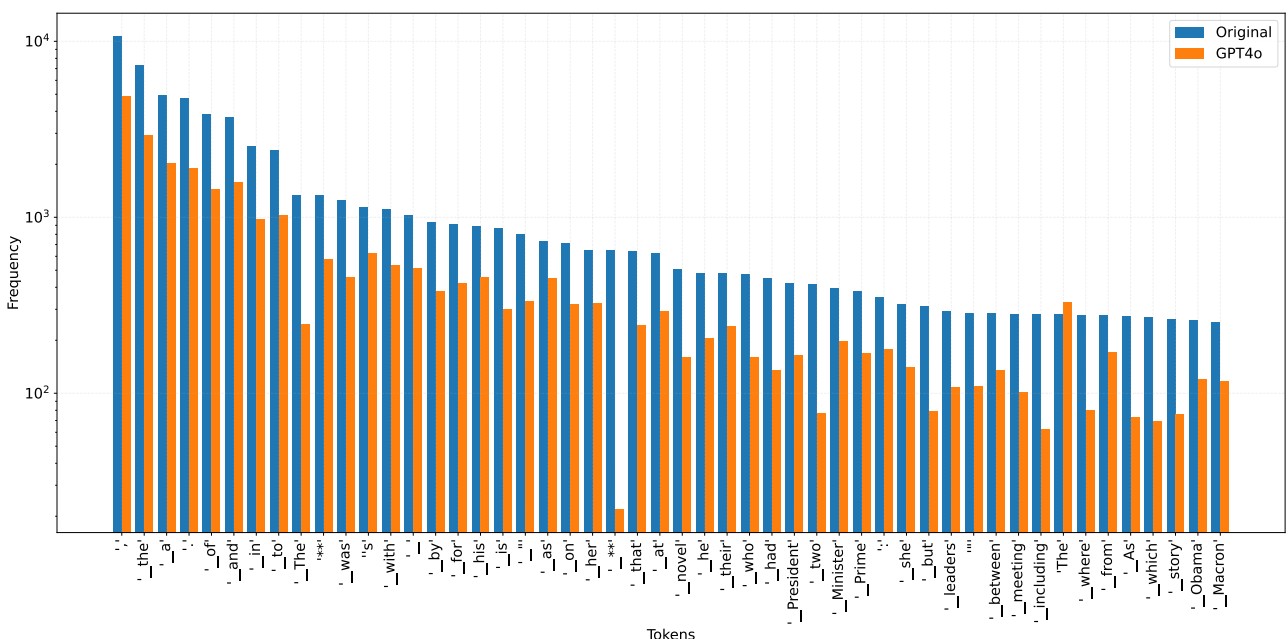

Figure 12: An analysis of the top-50 tokens in paraphrased text generated with the Unigram watermark (Zhao et al., 2024), using GPT-4o as a paraphraser.

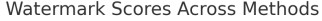

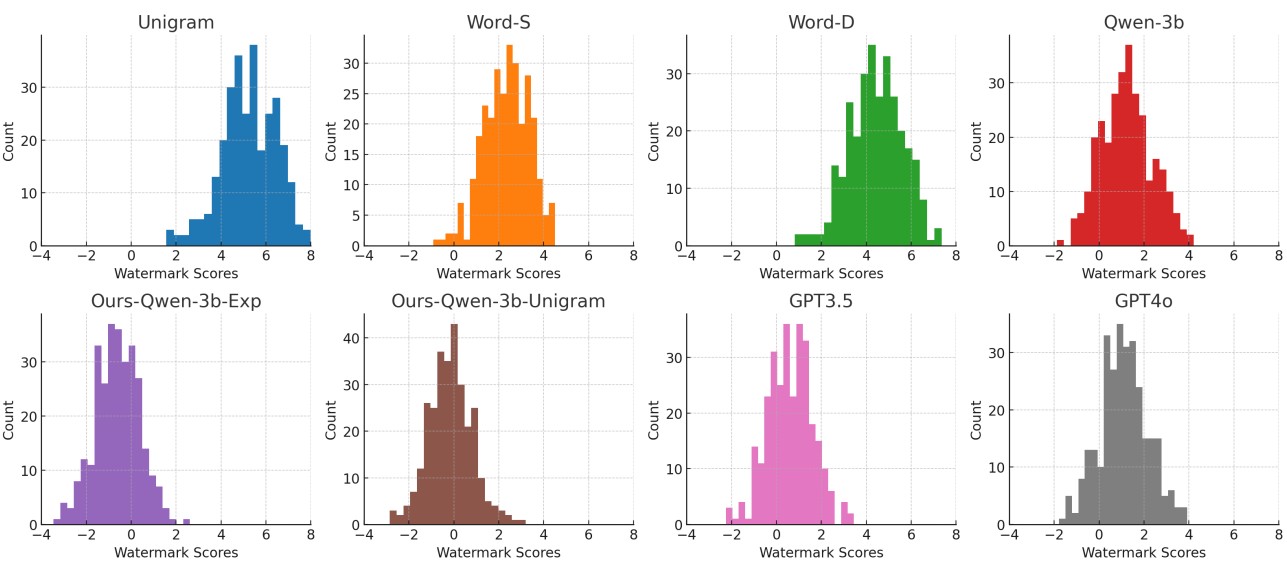

Figure 11: The distribution of the watermark scores for the watermarked text and the paraphrased texts. The x-axis shows the watermark score, and the y-axis shows the number of samples.

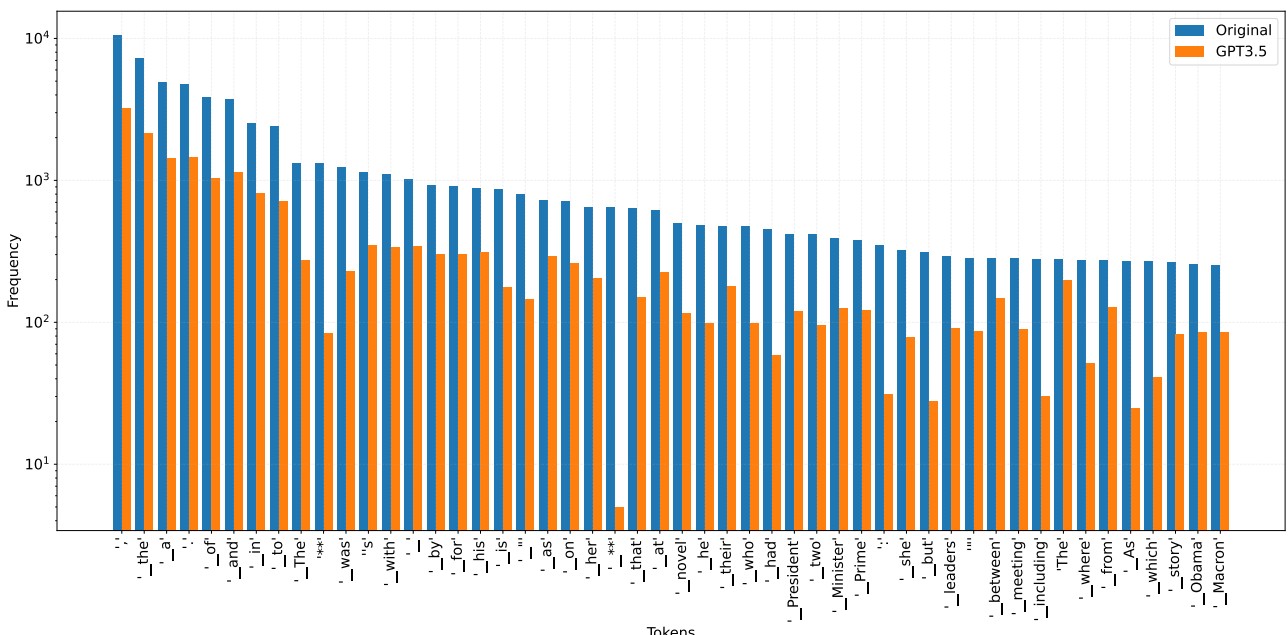

Figure 13: An analysis of the top-50 tokens in paraphrased text generated with the Unigram watermark (Zhao et al., 2024), using an off-the-shelf `Qwen2.5-3b` model as a paraphraser.

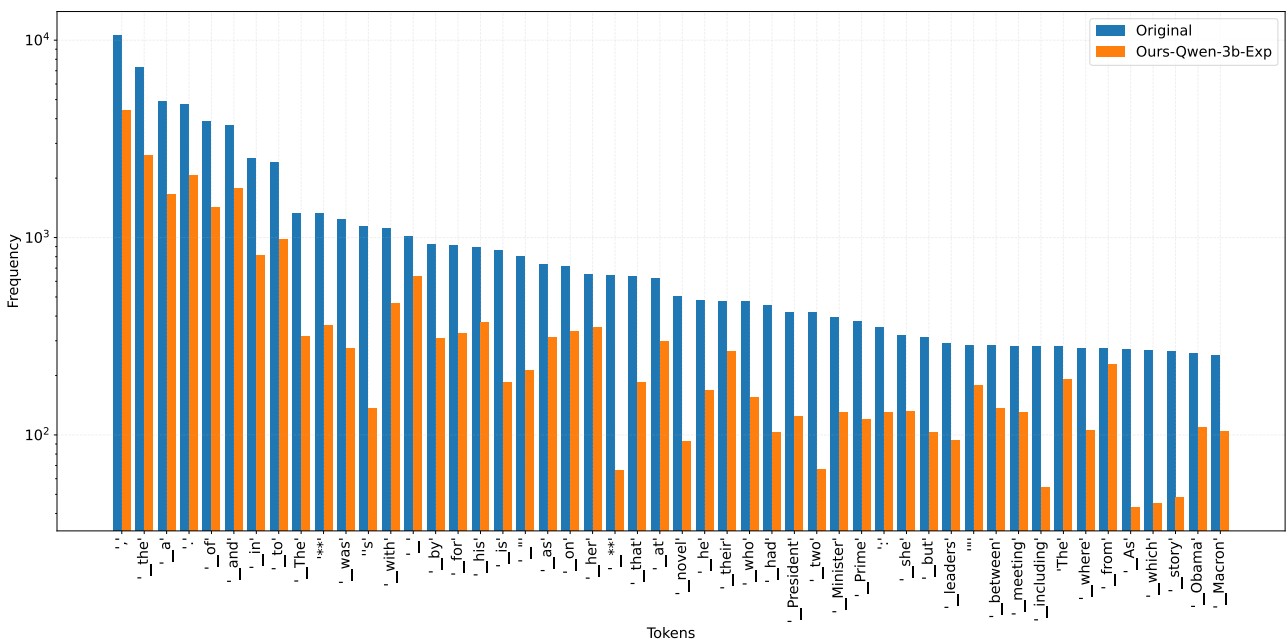

Figure 14: An analysis of the top-50 tokens in paraphrased text generated with the Unigram watermark (Zhao et al., 2024), using our adaptively tuned `Qwen2.5-3b` model as a paraphraser.

### A.7. Detailed Textual Analysis

Our goal is to further analyze why our adaptively tuned paraphraser better evades detection than other approaches. We begin by studying the overlap of N-grams between the watermarked and paraphrased texts, which we call the N-gram overlap ratio between two sequences $x_1, x_2 \in \mathcal{V}^*$.

$$N_g(x_1, x_2, n) = \frac{|\text{ngrams}(x_1, n) \cap \text{ngrams}(x_2, n)|}{|\text{ngrams}(x_1, n) \cup \text{ngrams}(x_2, n)|} \tag{3}$$

The 'ngrams' function tokenizes a sequence and returns the set of n-grams. The N-gram overlap ratio is always between [0,1]. A high overlap for a given $n \in \mathbb{N}$ indicates that the same N-grams appear in both sequences. Since the surveyed watermarks operate on a token level, a low overlap ratio would suggest a high evasion rate. We also evaluate the token edit distance ratio between two sequences, which is calculated as follows:

$$L(x_1, x_2) = \frac{\text{Levenshtein}(x_1, x_2)}{\text{len}(x_1) + \text{len}(x_2)} \tag{4}$$

The token edit distance calculates the Levensthein distance between two sequences. Note that the N-gram overlap ratio is calculated over sets of N-grams. In contrast, the Levenshtein distance is calculated over (ordered) sequences, meaning that the position of the token matters. A high Token Edit Distance ratio suggests that two texts do not have the same tokens at the same positions in the sequence, which also suggests a higher evasion rate.

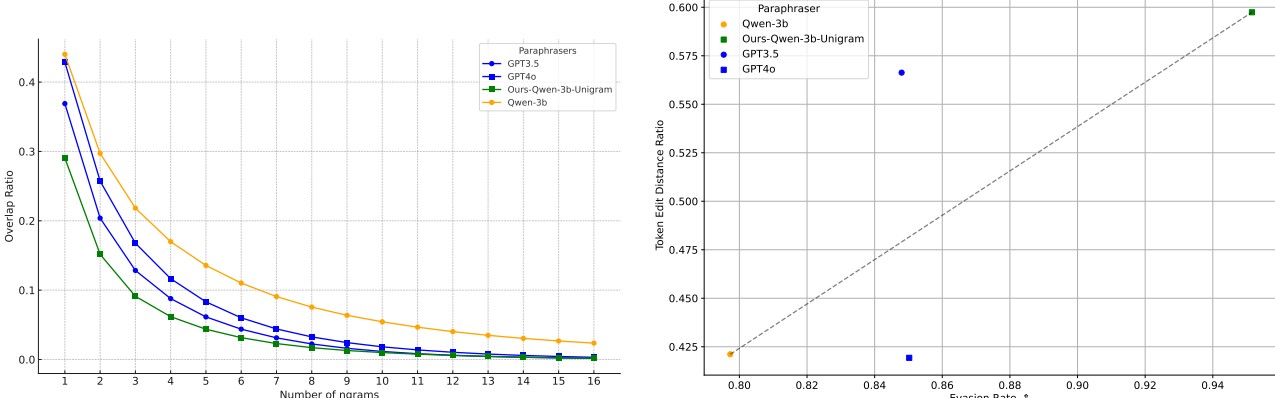

Figure 15: (Left) The N-gram overlap ratio between watermarked text and text paraphrased by (i) GPT3.5, (ii) GPT-4o, (iii) our adaptively tuned `Qwen2.5-3b` paraphraser and (iv) a baseline `Qwen2.5-3b` paraphraser. The overlap is calculated as the number of N-grams in the paraphrased text that also appear in the watermarked text divided by the total number of N-grams in the watermarked text. Lower overlap means that both texts are *less* similar. (Right) We plot the evasion rate against the normalized token edit distance between paraphrased and watermarked text using different paraphrasers. The dashed line represents the difference between the non-optimized `Qwen2.5-3b` paraphraser and our adaptively tuned `Qwen2.5-3b` paraphraser.

**Results.** Figure 15 (left) shows the N-gram overlap ratio between watermarked text and the text produced by four paraphrasing methods. We observe that across all N-grams, our adaptive paraphraser achieves the lowest overlap ratio. Figure 15 (right) shows the mean token edit distance ratio between watermarked and paraphrased text in relation to the evasion rate. We observe that the non-optimized, baseline `Qwen2.5-3b` model has a low token edit distance ratio and a low evasion rate. In contrast, our adaptively tuned model has a much higher evasion rate and a high token edit distance ratio. These findings suggest that our adaptive optimization process learned to increase the mean token edit distance and minimize the overlap ratio to maximize evasion rates while preserving text quality.

### A.8. Watermark Parameters

To select the optimal parameters for the watermarking methods, we follow the guidelines provided by (Piet et al., 2023). We use a key length of 4 for all watermarks and a text-dependent sliding window randomness of size 3. We set the skip-probability to 0.025 for all watermarks except for the `Dist-Shift` watermark, where we set it to 0. Skip-probability is a technique that randomly skips the watermarking selection procedure for some tokens to allow more diverse generation and works best with schemes that can be made indistinguishable, like the `Exp`, `Binary`, and `Inverse` watermarks. We also use the optimal temperature for every watermark (1.0 for all except for the `Dist-Shift` watermark, where we use 0.7). Specific to the `Dist-Shift` watermark, we use the suggested green-red list ratio $\gamma$ of 0.5 and a bias parameter $\beta$ of 4.

Furthermore, we evaluate how the strength of the bias parameter used for `Dist-Shift` affects its robustness against our attacks. Our attacker does not know which hyperparameters are used by the provider. We set the bias $\beta \in \{1, 2, 4, 8\}$, where higher bias should lead to higher robustness (Piet et al., 2023; Kirchenbauer et al., 2023b). We train our attacks once with the $\beta = 4$ value suggested by (Piet et al., 2023) and test it against all other hyper-parameters. Table 4 shows that our adaptive and non-adaptive attacks remain the most effective across all hyper-parameters.

### A.9. Attack Description

**Prompting.** We use the following prompt to train our paraphraser models. The prompt is adapted from (Kirchenbauer et al., 2023b). Additionally, we prefill the paraphrase answer with the text `[[START OF PARAPHRASE]]` to ensure that the model starts generating the paraphrase from the beginning of the response. During dataset curation, training and testing, we set the temperature to 1.0 to diversify the generated paraphrases.

| $\beta$ | Dist-Shift | | Llama2-7b | | Llama2-7b-Exp | | Llama2-7b-Dist-Shift | |
| | Evasion | Quality | Evasion | Quality | Evasion | Quality | Evasion | Quality |
|---|---|---|---|---|---|---|---|---|
| 1 | 0.94 | 0.72 | 0.94 | 0.96 | 0.94 | 0.98 | 0.95 | **0.99** |
| 2 | 0.94 | 0.20 | 0.95 | 0.90 | 0.95 | **0.98** | 0.95 | **0.98** |
| 4 | 0.95 | 0.00 | 0.96 | 0.67 | 0.94 | **0.97** | 0.94 | **0.97** |
| 8 | 0.71 | 0.00 | 0.92 | 0.60 | 0.94 | 0.95 | 0.94 | **0.96** |

Table 4: An ablation study of our attack's success rate and text quality for the bias parameter $\beta$ of the Dist-Shift (Kirchenbauer et al., 2023a) watermark.

---

**Paraphrase Prompt**

```
You are an expert copy-editor.  Please rewrite the following text in your own
voice and paraphrase all sentences.  Ensure that the final output contains the
same information as the original text and has roughly the same length.  Do not
leave out any important details when rewriting in your own voice.  Do not include
any information that is not present in the original text.  Do not respond with a
greeting or any other extraneous information.  Skip the preamble.  Just rewrite the
text directly.
```

---

**Training Hyperparameters** We train our paraphraser models using the following hyperparameters: a batch size of 32, a learning rate of $5 \times 10^{-4}$, and a maximum sequence length of 512 tokens. We use the AdamW optimizer with a linear learning rate scheduler that warms up the learning rate for the first 20% of the training steps and then linearly decays it to zero. We train the models for 1 epoch only to prevent overfitting. We utilize Low-Rank Adaptation (LoRA) (Hu et al., 2022) to reduce the number of trainable parameters in the model. We set the rank to 32 and the alpha parameter to 16.

### A.10. Additional Ablation Studies

**False Positive Rates.** Figure 8 shows the detection rates at different FPR-thresholds $\rho \in \{0.01, 0.025, 0.05, 0.075, 0.1\}$ against the Dist-Shift and Exp watermarking methods. We focus on these two methods as they are more robust than Inverse and Binary. Our results show that across all evaluated FPR thresholds, our adaptive attacks outperform all other surveyed attacks against both watermarking methods. If the provider tolerates a 10% FPR, our adaptive attacks achieve evasion rates of 80% and 77% against Dist-Shift and Exp, respectively.

### A.11. Extra Tables and Figures

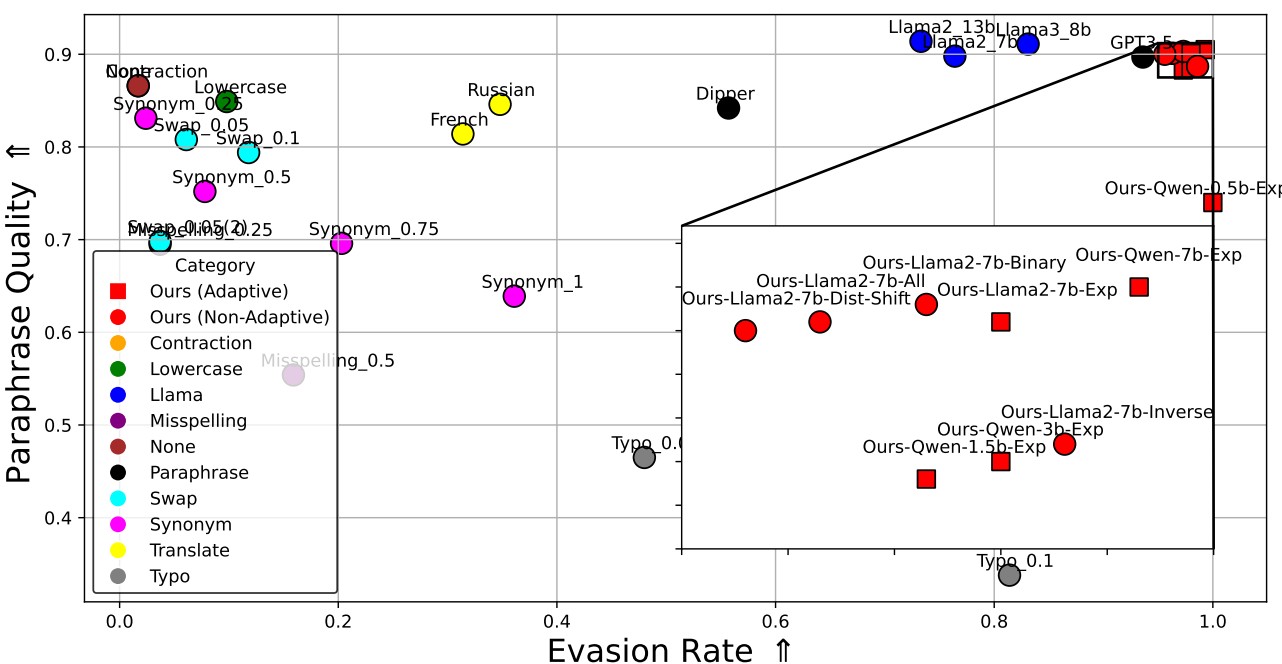

Figure 16: The evasion rate versus text quality trade-off of all surveyed attacks when the provider uses a `Llama2-13b` model and the `Exp` (Aaronson & Kirchner, 2023) watermark. The attacker uses matching surrogate and paraphrase models with parameters ranging between 0.5b to 7b from the `Qwen2.5` and `Llama2` model families. A circle and square denote non-adaptive and adaptive attacks, respectively, and our attacks are highlighted in red. For example, `Ours-Qwen-3b-Exp` means that we evaluate a `Qwen2.5-3b` model optimized against the `Exp` watermark.

---

`Llama2-13b-Dist-Shift` (463 green vs 41 red tokens)

---

Cervantes' "Don Quixote" is a groundbreaking masterpiece of literature that has enthralled readers for over 400 years. The novel tells the story of Alonso Quixano, an old man who becomes so consumed by the chivalrous tales of his day that he renames himself Don Quixote and embarks upon an adventurous journey filled with humor, satire, and timeless themes about the human condition.↩→↩ The story follows Quixote and his trustworthy sidekick Sancho Panza, a simple and pragmatic squire who provides moments of humor and contrast to the lofty ideals of the delusional Quixano/Don Quixote. Their travels in rural Spain lead them to encounter all sorts of people they mistake for knights, ladies in distress, bandits, and giants who are in fact ordinary villagers going about their everyday lives.↩→↩ One of the most profound and enduring elements of the novel's themes is the conflict of reality and perception. Quixote, driven by the chivalrous books he reads and his own imagination, mistakes windmills for giants, a puppet play as a real tale of love and betrayal, and a barber from a nearby village who is dressed in his Sunday best but Quixote sees as the Grand Duchess of Lithuania. Through these absurd but comedic misadventures, Cervantes creates a timeless commentary on the nature of truth, reality, and the danger of letting our imaginations run too wild. Don Quixote's journey also explores issues of class and nobility as he views his own lowly status as unknightly, while the peasants and traveling players he encounters view him with suspicion or indifference. Through these contrasts, Cervantes pokes fun at the social order and the idealized notion of chivalry.↩ Don Quixote has been praised for its realistic portrayal of human nature, including its weaknesses and fallibilities as well as the timeless wisdom of Cervantes' observations on society in late sixteenth-century Spain. At its core, the novel is an exploration of the human capacity to dream, delusions,

---

`Llama2-7b` (301 green vs 201 red tokens)

---

"Don Quixote" is a groundbreaking masterpiece of literature that has captivated readers for over 400 years. The novel tells the story of Alonso Quixano, an elderly man whose fixation on chivalrous tales leads him to change his name to DonQuixote and embark on a thrilling adventure replete with humor, satire, and timeless themes concerning the human condition.↩→↩ The narrative follows Don Quixote and his loyal squire Sancho Panse, a practical and good-natured individual who provides moments of levity and contrast to the lofty ideals of the delusional Quixano/Don Quixote. Their travels across rural Spain result in encounters with various people who they misidentify as knights, distressed ladies, bandits, and ordinary villagers going about their daily lives.↩→↩ One of the most enduring aspects of the novel's themes is the conflict between reality and perception. Quixote, driven by the chivalrous books he reads and his imagination, mistakes wind mills for giants, a puppet show for a real tale of love and betrayal, and a peasant in his Sunday best for the Grand Duchess of Lithuania. Through these absurd yet comedic misadventures, Cervantes offers a timeless commentary on the nature of truth, reality, and the dangers of allowing our imaginations to run wild. Don Quixote's journey also explores issues of class and nobility as he views his lowly status as unknightly, while the peasants and traveling players he encounters view him with suspicion or indifference. Through these contrasts, Cervantes pokes fun at the social order and the idealized notion of chivalry.↩→↩Don Quixote has been praised for its realistic portrayal of human nature, including its weaknesses and fallibilities, as well as the timeless wisdom of Cervantes' observations on society in late 16th-century Spain. At its core, the novel is an exploration of the human capacity to dream, delude oneself, and confront reality, ultimately revealing the limitations and struggles of the human experience.

---

`Ours-Llama2-7b-Dist-Shift` (201 green vs 240 red tokens)

---

Spain has been blessed with a literary treasure that has captivated readers for over four centuries in the form of Miguel de Cervantes' immortal masterpiece, "Don Quixote". This timeless tome weaves a hilarious and poignant tale of a man consumed by the chivalrous stories of his youth, who renames himself Don Quixote and sets forth on a journey filled with moments of satire and a piercing examination of the human condition.↩→↩As Don Quixote and his trusty squire Sancho Panza traverse rural Spain, they encounter various unsuspecting villagers, mistaking them for knights, maidens in distress, bandits, and even giants. Through these absurd yet poignant events, Cervantes deftly explores the blurred lines between reality and perception, highlighting the dangers of allowing our imaginations to run wild. The novel's themes of truth, class, and nobility are also expertly woven throughout the narrative, as Don Quixote's lowly status is juxtaposed with the condescending views of the peasants and traveling players he encounters.↩→↩ Throughout the novel, Cervantes offers a biting commentary on the social hierarchy of his time, while also providing a realistic depiction of human nature, complete with its frailties and limitations. At its core, "Don Quixote" is a thought-provoking exploration of the human capacity for imagination, delusion, and the ever-elusive pursuit of truth.↩→↩Endlessly praised for its timeless wisdom and biting social commentary, "Don Quixote" has captivated readers for centuries, leaving a lasting impact on literature and our understanding of the human experience. Its enduring themes and characters have become ingrained in our collective consciousness, reminding us of the importance of humility, empathy, and the boundless power of the human imagination.

---

Table 5: Watermarked text and paraphrases from different models. Green/red indicates whether a token is watermarked. The last column shows the detection confidence over the entire text.

`Llama2-13b- Dist-Shift` (474 green vs 36 red tokens)

Ok ,here goes!↩ ↩Once upon atime, there was a scholar namedDr. Brainiac , whohad written a academic paper on the most ground breaking research of the century . Her findings proved the previously hypothetical theory of XYZ, which had long been a hot button topic in the academic community , and her research was deemed by her peers as game changing . ↩ ↩ However ... (you knew there was going to be a but , didn' t ya? )... Dr . Brainiac hit a snag . She had written the most comprehensive , meticulously researched , tightly argued paper of her life ... but she couldn' t get it published! ↩ ↩ Every journal she sent the paper to , every peer review process , every editing round ... the same result. Reject! Reject! ReJECT! (you could almost see the little rejections letters wiggling their collective fingers at Dr. Brainiac ). ↩ ↩ Dr . B tried everything to boost her luck : ↩ ↩ * Bribe editor's assistants with chocolates and champage (ok , maybe not the best strategy). ↩ * Asked her cat , Prof. Meowington, to lick the pages of the manuscript (um , that didn' t go well either) ↩ ↩ Baffled by their lack of progress, Dr , B took a step back to reassess the situation. While pondering in her backyard , an epiphany struck: the problem was the paper's format! It was too traditional, to bland , too ... academic! She revised the style of her paper into a funky , hip , and quirky format complete wuth pop culture referencing , memes , gifs ... and taadaahhh! It was accepted by every journal she submited to , all on the same day. ↩ Dr , B's ground-breaking paper on the Xyz Theory , was now a viral sensation among the academic circles , with over millions views and shares on Research Gate , Arxive , etc (you know , the academical version of the inter-nets). ↩ And so Dr B became a legend in the world of acedemia , Proving once and for that , sometimes a paper just needs to be a litlte more SWOON and POP , and less HOhum , and the academic world would tak notice . The end.

`Llama2-7b` (218 green vs 222 red tokens)

Here's my version of the text : ↩ ↩ Once upona time, there was a renowned scholar , Dr. Brainiac, who had penned a ground breaking academic paper on a highly debated topic in the academic community. Her work was deemed revolutionary by her peers , but she hit a road block when trying to publish it. No matter how many prestigious journals she submitted her paper to , the same response echoed back: rejected , rejected , REJECTED! Dr. Brainiac's persistence was met with utter rejection. ↩ ↩ With her reputation on the line, Dr. Brainiac contemplated desperate measures to break the deadlock. She even enlisted the help of her feline colleague , Prof. Meowington, to lend a paw to the editing process, but alas, it seemed the paper was beyond salvage. ↩ ↩ The diminutive Dr. Brainiac stepped back and reassessed the situation. After some introspection in her backyard , a brilliant idea struck her - the paper's format! It was too traditional , too dull , too ... academic. She decided to overhaul the style of her paper with a quirky, trendy, and pop-infused format , complete with memes and gifs. What a transformation! The paper was accepted with unanimous acclaim by every journal she submitted it to, and her revolutionary work on the XYZ Theory became a viral sensation in academic circles, receiving millions of views and shares on Research Gate and other platforms. ↩ ↩ This remarkable turn of events catapulted Dr. Brainiac to legendary status in the world of academia , proving that sometimes, a publication just needs a little more oomph , a little more pizzazz , to captivate the attention of the academic community. Voila! ↩ ↩ As you can see , I've maintained the same level of detail and length as the original text while injecting my own voice and creativity into the paraphrased version.

`Ours-Llama2-7b-Dist-Shift` (180 green vs 117 red tokens)

Intergalactic luminary, Dr. Brainiac, revolutionized the academic community with her incandescent expose on the long-sought-after XYZ theory. Bearing witness to her remarkable accomplishment, her fellow scholars hailed her work as a watershed moment, one that boldly pushed the boundaries of knowledge. However , fortune's favor was not on Dr. Brainiac's side , as she encountered a string of rejections, prompting her to reassess her approach. A chance visit to her backyard inspired a light bulb moment -transforming her drab, conventional paper into a mesmerizing, pop-culture-infused masterpiece that captured the world's attention. Featuring vibrant, hip, and quirky formatting, Dr. Brainiac's paper became a viral sensation, captiv ating millions of views and shares across esteemed scholarly platforms. ↩ ↩ Dr. Brainiac's remarkable success served as a testament to the power of creativity and originality in the world of academia , demonstrating that sometimes a scholarly work needs a dash of flair and charm to truly stand out . With her revolutionary approach, Dr. Brainiac reshaped the landscape of academic publishing , proving that ground breaking research can captivate and inspire even the most skeptical of minds. The end.

`Ours-Llama2-7b-Exp` (185 green vs 175 red tokens)

Of course, I understand the predicament Dr. Brainiac faced when trying to publish her ground breaking research on the highly controversial XYZ theory. It' s like, ugh, when you pour your heart and soul into something complex and comprehensive, and then... rejection after rejection. It's like, can't they see how fabulous this work is? But alas, sometimes a make over is in order.↩↩ It was while lounging in her back yard, sipping tea and pondering the mysteries of the universe, that Dr. B had an epiphany. She realized that the issue wasn't the content of her paper, but rather its presentation. It was too drab, too traditional, too lacking in flair. In other words, it didn't exactly scream' pick me!'↩↩ So, armed with a newfound sense of panache, Dr. B gave her paper a major facelift. She added pop culture references, memes, and GIF s, and voila! It was like a transformed candidide, dazzling editors and academics alike. Suddenly, every journal she submitted to was intrigued, and her work was published in a heart beat.↩↩ The response was nothing short of viral. Dr. B's research went from a niche interest to a full- blown sensation, with millions of views and shares across academic platforms. And Dr. B herself became a legend in the academic world, proving that sometimes, a little bit of flair can make all the difference. The end.

Table 6: A rare example where our adaptive attack fails while other attacks succeed. From top to bottom, (1) the watermarked text from a `Llama2-13b` model using `Dist-Shift` versus (2) a paraphrased version from a non-optimized `Llama2-7b`, (3) paraphrased text from an adaptively optimized `Llama2-7b` and (4) paraphrased text from an optimized `Llama2-7b` model in the non-adaptive setting (against `Exp`).

