# OpenReview forum: "Optimizing Adaptive Attacks against Watermarks for Language Models"
_ICML.cc/2025/Conference — ICML 2025 spotlightposter_

### Official Review · Reviewer_3sCX · 2025-02-24

**Overall Recommendation:** 4

**Summary:**

This work proposes adaptive paraphrasing models as a new attack vector against various LLM watermarks. The adversary primarily targets an adaptive no-box setting, where the watermarking algorithm (but not keys, etc.) is known, but the adversary has no access to the watermarking model itself (e.g., for querying). In the first step, by using a surrogate model with the same type of watermark (or, in some instances, a mixture), the adversary creates a preference tuning (DPO) dataset. On this, the adversary RL finetunes a paraphrasing model, which will then be used to remove the watermarked text received from the target model. This is evaluated across several currently popular watermarking schemes showing consistently higher watermark removal rates (with generally high quality). The work highlights the need for stronger adversarial evaluation of LLM watermark methods.

## Update after rebuttal

The reviewer stands by their decision at the end of the reviewer rebuttal discussion. The rebuttal successfully addressed most of my concerns, and as such, I favor acceptance.

**Claims And Evidence:**

Yes all major claims regarding evasion rate and quality preservation are sufficiently backed up by experimental evidence.

**Essential References Not Discussed:**

None.

**Experimental Designs Or Analyses:**

Warranted the notes in "Methods And Evaluation Criteria", the design seems sound to the reviewer.

**Methods And Evaluation Criteria:**

The reviewer checked the experimental design wherever possible. The reviewer could not find a description of the dataset which was used for the final evaluation. In case it is the same dataset as the dataset used for RL finetuning this may have potential consequences for the validity of some claims as it potentially leaks information.

**Other Comments Or Suggestions:**

- To the reviewer's understanding, Pareto optimal is really a theoretical statement (which one would have to make overall existing adversaries) - what can be surely stated is that their method is on the current Pareto-frontier (but one should probably refrain from calling it optimal).
- The notion of including the watermark directly in the model parameters (L55R) is somewhat of a strange choice as there exists an entire field of Open-Source watermarking that specializes on this [2] (containing the watermark directly in the model). As this is not the focus of this work, it may help the presentation if it is slightly revised.

[2] Sander, Tom et al. "Watermarking Makes Language Models Radioactive." _ArXiv_ abs/2402.14904 (2024): n. pag.

**Other Strengths And Weaknesses:**

#### Strengths
- The reviewer enjoyed reading the paper and thinks the core idea is sensible. Further stronger attacks and evaluations of LM Watermarking is a timely effort with their increasing popularity.
- The evaluation is thorough with (mostly) realistic attacker assumptions containing various attack targets and many baselines (with varying architectures) alongside quality evaluations.
- The method shows strong overall results - especially the transfer attacks
#### Weaknesses

- The reviewer appreciates the additional explanations as to why the paraphrasing seems so effective but has to note that the provided explanations are largely empirical in nature, leaving some uncertainty about where to go from here for future, more robust, watermarks.
- The phrasing "existing work only exists against non-adaptive attackers" (e.g., in the abstract) does not seem to do justice to existing work that (while not in a no-box setting) does make some assumptions about the watermark under attack and adapts/learns based on this [1].
- Zero-shot baselines already seem very strong, which partially diminishes the gains of the methods (this is a minor point as the reviewer agrees that the method requires much smaller models) -> However, it raises the question of how far one could push a zero-shot LLM, especially with some additional filters/modifications.
- The quality results across the main paper are on a quite weak judge model - the results in A.3 are hard to compare in put in reference but show consistently lower scores when using a stronger judge model. This could hint at a high variance depending on the quality score. Related to this, a full overview of the score distributions in Table 3 could help people contextualize the score distributions.

[1] Jovanović, Nikola, Robin Staab, and Martin Vechev. "Watermark stealing in large language models." ICML (2024).

**Questions For Authors:**

- The provided examples seem to show that paraphrases by the method are shorter. Can the authors provide additional statistics on this and potential reasons (e.g., from the preference dataset)?
- On which dataset was the evaluation conducted (number and source of samples, distribution, etc.)?
- See other points above.

**Relation To Broader Scientific Literature:**

The work provides a new angle on the evaluation of the robustness of current LLM watermarking schemes. Most prior work primarily focussed on standard paraphrasing, basic character replacement, or human evaluation of watermark robustness (with some exceptions as outlined in other sections). This work is novel in the sense that it not only uses paraphrasers but specifically finetunes them to remove watermarks from the text. This becomes an increasingly realistic threat model under the more widespread deployment of LLM watermarks.

**Theoretical Claims:**

The work does not include theoretical claims.

---

> ### Author Rebuttal · Authors · 2025-03-30
>
> Thank you for your time, valuable suggestions, and excitement about our paper. Please find our responses below.
>
> > [..] explanations are largely empirical in nature, leaving some uncertainty about where to go from here for future, more robust, watermarks.
>
> Thank you for raising this point, which was also raised by other reviewers. We discuss future work in Section 6, where we propose 'adversarial training', which would first require designing optimizable defenses against our optimizable attacks. This is not a trivial task, hence we believe this is future work. We kindly refer to our response to reviewer 'z2QJ' for more information on 'where to go from here'.
>
> > The phrasing "existing work only exists against non-adaptive attackers" (e.g., in the abstract) does not seem to do justice to existing work that (while not in a no-box setting) does make some assumptions about the watermark under attack and adapts/learns based on this [1].
>
> In the abstract, we claim that "[..] robustness is tested only against non-adaptive attackers" which we believe is valid for all the surveyed watermarking methods. We do not claim that we are the first adaptive attack. While Jovanović et al. [1] do not explicitly use the term 'adaptive' in their paper, we agree with the reviewer that their spoofing attack works only if they know the watermarking method (i.e., they adaptively handcrafted their attack). We will revise the paper to clarify this.
>
> > Zero-shot baselines already seem very strong [..] However, it raises the question of how far one could push a zero-shot LLM, especially with some additional filters/modifications.
>
> Thank you for raising this point. We also think this would be an interesting question. One can use our attack method to adaptively optimize for such a prompt (i.e., treat the prompt as the optimizable parameters), which we did not do in our paper. We already used handcrafted prompts that we manually optimized, which likely explains the strong zero-shot baselines.
>
> > [..] Related to this, a full overview of the score distributions in Table 3 could help people contextualize the score distributions.
>
> This is a good idea, and we thank the reviewer for bringing it up. We will revise the paper to include a histogram of GPT-Judge scores in the Appendix.
>
> > To the reviewer's understanding, Pareto optimal is really a theoretical statement (which one would have to make overall existing adversaries) - what can be surely stated is that their method is on the current Pareto-frontier (but one should probably refrain from calling it optimal).
>
> We are unsure if we correctly understood what the reviewer refers to when they say that 'Pareto optimal is really a theoretical statement' and would kindly ask for further clarification in case our answer missed the point. Given all other surveyed attacks, our attacks are Pareto-optimal. However, as the reviewer points out, there could be even better attacks that we did not evaluate.
>
> > The notion of including the watermark directly in the model parameters (L55R) is somewhat of a strange choice as there exists an entire field of Open-Source watermarking that specializes on this [2] (containing the watermark directly in the model). As this is not the focus of this work, it may help the presentation if it is slightly revised.
>
> We agree with the reviewer that this might be confusing, as the watermarks we consider do not necessarily modify the model's parameters. We will clarify that $\operatorname{Embed}$ can change the entire generation procedure (such as warping the logits of the LLM during generation).
>
> > The provided examples seem to show that paraphrases by the method are shorter. Can the authors provide additional statistics on this and potential reasons (e.g., from the preference dataset)?
>
> Yes, this is likely an effect of the optimization, which does not explicitly penalize shorter responses. This property of the output could be controlled by modifying the objective function. We evaluated our data and observe that a non-optimized baseline Qwen2.5-3b has a mean output token length after paraphrasing of $\mu=230$ tokens ($\sigma=25.25$), whereas our optimized version has an output length of $\hat{\mu}=214$ tokens ($\hat{\sigma}=33.16$). We will include these results in the revised paper.
>
> > On which dataset was the evaluation conducted (number and source of samples, distribution, etc.)?
>
> We follow Piet et al. [A] 's evaluation and kindly refer the reviewer to their paper for more details. The number of samples is 296 across different tasks (storylines, reports, fake news prompts, etc.). We will also add a short section describing the dataset in the Appendix to ensure our paper remains self-contained.
>
> ---
> [A] Piet, Julien, et al. "Mark my words: Analyzing and evaluating language model watermarks." arXiv preprint arXiv:2312.00273 (2023).

---

> > ### Comment · Reviewer_3sCX · 2025-04-02
> >
> > I thank the authors for their rebuttal and am happy to see the corresponding changes are to be included in the manuscript. Overall, the reviewer maintains their favorable view and thinks this is a valuable contribution to the field - as such, I will raise my score to favor acceptance.

---

### Official Review · Reviewer_zipU · 2025-03-08

**Overall Recommendation:** 3

**Summary:**

The paper proposes to apply preference optimization (DPO) to make open LLM paraphrasers better at removing watermarks from texts. The focus is on the nobox setting (the attacker has no previous access to the LLM API) and both adaptive (scheme is known but not its key) and non-adaptive settings (the exact scheme is not known).

## Update after rebuttal

As discussed with authors below the rebuttal has addressed most of my concerns, so I raise my score to favor acceptance.

**Claims And Evidence:**

Claims are generally supported by evidence. However, given that non-adaptive attacks seem to work comparably well, to me the title of the paper and the key pitch in the abstract/introduction set up wrong expectations, as they are all centered around prior work not taking into account adaptive attacks while this paper does. To me, given that adaptive/non-adaptive both work well, the unique focus of this paper compared to prior work is the *nobox* setting and key message of the paper is that training against _any_ watermark can improve removal success against the target watermark.

**Essential References Not Discussed:**

No specific works are missing, but given the specific positioning of the paper, the related work section could come early instead of at the end, to clearly delineate the space in which the paper operates, and explain that existing blackbox attacks are not directly comparable.

Detail: in the introduction the citation Nicks 2024 for adaptive attacks is work that does not target watermarks at all (but other LLM text detectors). The other citation is about images, which is strange given that there is a line of work on adaptive (albeit blackbox) attacks on LLMs, some of which are cited in the related work section.

**Experimental Designs Or Analyses:**

I checked soundness of experiments and did not find issues. One detail I was unable to verify is the relationship between the dataset used for evaluation (custom questions as noted in A.2) and the dataset used to fine-tune the model. If these are the same/very similar this could imply leakage and perhaps the paraphrasers would not be as effective outside of this domain?

**Methods And Evaluation Criteria:**

Methods/evaluation criteria generally make sense. Given that it seems that nobox is the key feature of the proposed setting, the authors could elaborate more on the motivation for this setting. In particular, in which scenario does an attacker aim to remove a watermark from an LLM deployment but does not have prior blackbox access to that deployment? This would greatly help make the case for the paper.

**Other Comments Or Suggestions:**

- Figure 5 seems to have blue and green colors swapped

**Other Strengths And Weaknesses:**

(+) Generally, the paper is interesting and valuable to the community as it evaluates a reasonable idea that many practictioners might be interested in experimenting with

(+) The "why non-adaptive attacks would work" part is very interesting.

(-) There are several strange points in the evaluation:
 - The primary pareto front figure (Fig. 4) uses only weak/outdated closed models (e.g. gpt3.5) and models such as gpt4o are only shown in the appendix. As the key contribution of the paper in my opinion hinges on the method clearly beating off-the-shelf methods this is strange, and I would have expected a range of latest closed models in the main evaluation results.
 - Even weak off-the-shelf models generally get high scores (>80%) which does not match my intuition from prior work, where in difficult settings off-the-shelf paraphrasing is very weak. Perhaps including an evaluation with longer texts would be useful? Significantly improving a case where off-the-shelf models get ~20% would make a very strong case for the method compared to improvements in the 80-100 range.

Summarizing all of above, I appreciate the work and believe it can be valuable for the community but have key concerns around: paper phrasing around adaptiveness instead of nobox, motivation for nobox in particular, the choice of datasets, and the evaluation setting (latest off-the-shelf models and setting difficulty).

**Questions For Authors:**

- The $(\epsilon, \delta)$ robustness definition is interesting, but I have not seen it used later, why is htis needed?
- What are the "messages" $m$ here given that all methods are zero-bit to the best of my understanding?
- When mining negative samples how many of those fail the quality criterion and how many simply don't remove the watermark (and how many fail both criteria)? This analysis would be interesting. If this is imbalanced do you think balancing it on purpose could improve the results further?

**Relation To Broader Scientific Literature:**

As discussed above and below, the paper is a solid contribution and novel in the specific setting carved out.

**Theoretical Claims:**

/

---

> ### Author Rebuttal · Authors · 2025-03-30
>
> Thank you for the detailed response and generally positive outlook on our paper. We appreciate your time and effort. Please find our responses below.
>
> > [..] key message of the paper is that training against *any* watermark can improve removal success against the target watermark.
>
> We agree with the reviewer that all watermarks are vulnerable to our optimizable attacks in the non-adaptive setting. Our non-adaptive attacks differ from existing ones in that we *adaptively optimize* them against other (similar) watermarks and then show in our experiments their effectiveness transfers to unseen watermarks (likely due to their similarity; see Section 6). Hence, even though we evaluate these attacks in a non-adaptive setting, they were still *adaptively* tuned against other known watermarks.
>
> We believe these are core findings of our paper and will follow the reviewer's suggestion to highlight that 'training against *any* watermark can improve removal success against the target watermark'. However, we would also like to emphasize that in the future, more robust watermarks may be developed that resist our attacks in the non-adaptive setting but not in the adaptive setting.
>
> > [..] in which scenario does an attacker aim to remove a watermark from an LLM deployment but does not have prior blackbox access to that deployment?
>
> We agree with the reviewer that all deployed watermarking methods allow users to generate (some) watermarked samples. Realistic attackers typically have at least black-box access to the LLM and would not be confined to the no-box setting.
>
> Our motivation is to evaluate the robustness of watermarking against *constrained* attackers that (i) have limited resources and (ii) lack *any* information about the watermarking key and samples. If successful attacks exist in this pessimistic no-box setting, the provider cannot hope to have a robust watermark against more capable attackers (e.g., with black-box access). We show that (i) such attacks exist, (ii) they are cheap, and (iii) they do not require access to watermarked samples. We believe the development of defenses should focus on the no-box setting first. We will discuss this in the revised paper.
>
> > [..] If these [the datasets] are the same/very similar this could imply leakage and perhaps the paraphrasers would not be as effective outside of this domain?
>
> We used different query datasets to train and test the paraphraser and will clarify this in the paper. However, we highlight that in a practical setting, it is not unreasonable to assume that the attacker uses the *same* dataset to train and test the paraphraser. Since our attacks are cheap ($\leq10$$ USD), the attacker can train such a specialized paraphraser.
>
> > The (ϵ,δ) robustness definition is interesting, but I have not seen it used later, why is htis needed?
>
> We re-use $\delta$ in Alg. 2 and Section 4.2, and $\epsilon$ corresponds to the evasion rate, which we show in Figures 2, 3, 4. From these figures, one can deduce the $(\epsilon, \delta)$-robustness of the surveyed watermarks, subject to the metrics we study. We will clarify this in the paper.
>
> > What are the "messages" m here given that all methods are zero-bit to the best of my understanding?
>
> The reviewer is correct that all watermarks we focus on in the papers are zero-bit. Our definition in Section 2 is more general and allows for multi-bit watermarks as defined by Zhao et al. [A].
>
> > Even weak off-the-shelf models generally get high scores (>80%) which does not match my intuition from prior work [..]
>
> To enhance evasion rates, we have manually optimized the prompt used for the baseline models (page 18 in the Appendix).
>
> > When mining negative samples how many of those fail the quality criterion and how many simply don't remove the watermark (and how many fail both criteria)? [..] do you think balancing it on purpose could improve the results further?
>
> That is an interesting idea. To answer your question, we investigated the samples collected for training the paraphrasers on Unigram for Llama-3.2-3B. High quality means LLM-Judge had a score of $\geq 0.8$.
>
> |                 | High Quality | Low Quality | Total |
> |-----------------|------------:|------------:|------:|
> | Detected        | 28.21% | 9.99% | 38.20% |
> | Not Detected    | 48.14% | 13.66% | 61.80% |
> | Total           | 76.35% | 23.65% | 100.00% |
>
> We did not investigate methods to 'balance' negative samples since our current approach is already highly effective. This would be an interesting idea for further optimization.
>
> We will follow the reviewer's suggestions for the remaining points: (i) The citations in the introduction, (ii) add GPT4o to Figure 4 (it performs similarly to GPT3.5 in terms of evasion rate), and (iii) move the related work section to an earlier part of the paper. We believe the colors for Figure 5 are correct.
>
> ---
> [A] Zhao, Xuandong, et al. "SoK: Watermarking for AI-Generated Content." arXiv preprint arXiv:2411.18479 (2024).

---

> > ### Comment · Reviewer_zipU · 2025-04-04
> >
> > The authors have addressed most of my concerns, so I am raising to favor acceptance.
> >
> > We initially disagreed on the meaning "adaptive": I thought what makes the attacker adaptive is the knowledge of the particular watermark, but for the authors its about the knowledge /that there is a watermark/. This is reasonable but might be worth making it more explicit in the paper to avoid this mismatch.
> >
> > I believe every paper tries to optimize the paraphraser prompt, so this does not seem to me like a strong enough argument for the baseline methods performing better here than in some of the prior works; I would still find the results more convincing if there was a setting of longer texts included too.
> >
> > Figure 5 caption says: "Dist-Shift watermark (blue)" but the figure legend says shows Dist-Shift as green, so I insist that there is a typo here, but of course this does not affect my evaluation.

---

> > > ### Author Response · Authors · 2025-04-08
> > >
> > > Thank you for your time, positive assessment of our work, and support for its acceptance. As promised in our rebuttal, we will revise the paper to clarify precisely what we mean by adaptive and address all other points.
> > >
> > > In Figure 5, the reviewer is correct that the caption has a typo, which we will fix.

---

### Official Review · Reviewer_Q7gq · 2025-03-10

**Overall Recommendation:** 2

**Summary:**

The paper investigates the robustness of Large Language Model (LLM) watermarking. While previous research has primarily tested watermarking against non-adaptive attackers (lack knowledge of the watermarking technique), this study introduces an approach by formulating robustness as an objective function and using preference-based optimization (DPO) to develop adaptive attacks. The evaluation reveals three findings: (i) adaptive attacks effectively evade detection across all surveyed watermarking methods, (ii) optimization-based attacks, once trained on known watermarks, generalize well to unseen watermarks even in non-adaptive settings, and (iii) these attacks are computationally efficient, requiring less than seven GPU hours.

## update after rebuttal
Thanks for the rebuttal and the additional experiments. I remain unconvinced that pursuing stronger attacks is the right direction for advancing watermarking, and I am maintaining my original score. However, I am not opposed to the paper being accepted.

**Claims And Evidence:**

The claims are validated by the proposed adaptive attack algorithm and supported by the experimental results.

**Essential References Not Discussed:**

[1] Zhang, Hanlin, Benjamin L. Edelman, Danilo Francati, Daniele Venturi, Giuseppe Ateniese, and Boaz Barak. "Watermarks in the sand: Impossibility of strong watermarking for generative models." arXiv preprint arXiv:2311.04378 (2023).
https://hanlin-zhang.com/impossibility-watermarks/

[2] Fairoze, Jaiden, Sanjam Garg, Somesh Jha, Saeed Mahloujifar, Mohammad Mahmoody, and Mingyuan Wang. "Publicly-detectable watermarking for language models." arXiv preprint arXiv:2310.18491 (2023).

[3] Liu, Yepeng, and Yuheng Bu. "Adaptive Text Watermark for Large Language Models." In International Conference on Machine Learning, pp. 30718-30737. PMLR, 2024.

Discussions see Weaknesses.

**Experimental Designs Or Analyses:**

Yes, I checked all the experimental designs or analyses in the main body of the paper.

**Methods And Evaluation Criteria:**

Yes, the methods and evaluation are reasonable. However, the main body primarily relies on LLM-Judge for quality evaluation, while the perplexity performance, as shown in Table 3, is not particularly strong.

**Other Comments Or Suggestions:**

N/A

**Other Strengths And Weaknesses:**

This paper introduces a strong attack targeting the robustness of LLM watermarking methods. However, reference [1] **proves** that achieving strong watermarking is fundamentally **impossible**, as a watermarking algorithm cannot embed watermarks in all high-quality sentences while simultaneously ensuring they are recognized as LLM-generated.

The core idea behind their proposed attack is that it is always possible to find a high-quality response that differs significantly from the watermarked output. The proposed method appears to be a specific case of the attack discussed in the following paper, incorporating optimization and DPO.

Due to this inherent impossibility, most recent theoretical research on watermarking has shifted toward the notion of weak robustness, i.e., change in few tokens. See Definition 2.10 in [2].

Therefore, I disagree with the authors' claim that investigating strong attacks is necessary, as defending them is nearly impossible. Instead, to provide meaningful guarantees, it is more practical to formulate the problem by restricting the attacker's capabilities (e.g., a college student trying to submit an AI-generated essay) rather than assuming an adversary familiar with LLM watermarking.

**Questions For Authors:**

1. Conceptually, can the proposed attack be viewed as a specific instance of the one studied in [1]? How should I position this work given their impossibility result?

2. Reference [3] introduces a more robust semantic-based watermarking method compared to SIR. Is the proposed attack still effective against this approach? A new plot similar to Figure 7 should be included to illustrate the results.

**Relation To Broader Scientific Literature:**

See weakness.

**Theoretical Claims:**

This is an empirical paper without any theoretical claims.

---

> ### Author Rebuttal · Authors · 2025-03-30
>
> Thank you for your time, consideration, and many valuable suggestions. Please find our responses below.
>
> > Yes, the methods and evaluation are reasonable. However, the main body primarily relies on LLM-Judge for quality evaluation, while the perplexity performance, as shown in Table 3, is not particularly strong.
>
> Perplexity is indeed worse than the baseline model. However, as noted in the paper, perplexity is not always a good quality indicator since it penalizes diverse outputs. For this reason, we include many quality metrics (LLM-Judge, LLM-CoT, LLM-Compare, Mauve, and Perplexity) as described in Appendix A.1. These metrics show that our paraphrased text has a high quality.
>
> > [..] reference [1] proves that achieving strong watermarking is fundamentally impossible [..] I disagree with the authors' claim that investigating strong attacks is necessary, as defending them is nearly impossible. Instead, to provide meaningful guarantees, it is more practical to formulate the problem by restricting the attacker's capabilities (e.g., a college student trying to submit an AI-generated essay) rather than assuming an adversary familiar with LLM watermarking.
>
> Thank you for bringing up the impossibility results for 'strong' watermarking. We agree with the reviewer, and our primary setting is studying a resource-constrained attacker - but we are interested in the strongest possible such attacker. Staying with their analogy of viewing watermark evasion as graphs, the attacks in [1] use random walks, whereas our attacks are optimized to find the shortest paths. This makes our attacks more computationally efficient and allows the use of smaller paraphrasers than [1].
>
> We already assume a particularly constrained attacker who (i) has no access to the provider's watermarked LLM and (ii) has limited computational resources. Strong watermarking is fundamentally impossible (if quality and perturbation oracles exist), which makes it interesting to study whether watermarking exists that is robust in practice against constrained attackers (i.e., evasion would incur prohibitively high costs). Our work studies this setting and proposes effective attacks that incur little computational costs to the attacker of less than $10$$ USD.
>
> > Conceptually, can the proposed attack be viewed as a specific instance of the one studied in [1]? How should I position this work given their impossibility result?
>
> As described above, the attack in [1] uses a random walk and proves that under certain assumptions, this theoretically guarantees removal, irrespective of the computational costs incurred for the attacker. Since we optimize our paraphraser, we have a higher probability of evading detection with fewer steps (1 step in our work). We could additionally control other properties of the output (e.g., semantic similarity to the original watermarked text). We will add this discussion to the revised paper.
>
> > Reference [3] introduces a more robust semantic-based watermarking method compared to SIR. Is the proposed attack still effective against this approach? A new plot similar to Figure 7 should be included to illustrate the results.
>
> Thank you for bringing [3] to our attention. We evaluated it against our pre-trained models and obtained results similar to other watermarks we surveyed. We use the provider model Llama-2-13b. Note that $\epsilon$ refers to the evasion rate, as described in our paper. Below, we use watermarking strengths $(\delta_0=0.33, \delta_1=0.33)$ (not to confuse with our notation of $\delta$ for text quality)
>
> | Method                                       |   $\epsilon$ |   GPT Judge Rating |
> |:---------------------------------------------|------:|-------------------:|
> | Watermarked Samples | 3.72 |               0.71 |
> | Qwen/Qwen2.5-3B-Instruct | 92.23 |               0.72 |
> | meta-llama/Llama-3.1-8B-Instruct | 98.65 |               0.74 |
> | Ours-Unigram-Qwen2.5-3B | 98.65 |               0.74 |
> | Ours-Unigram-Llama-3.2-3B | 100.00 |               0.77 |
> | Ours-EXP-Qwen2.5-3B | 99.32 |               0.71 |
> | Ours-KGW-Llama-2-7B | 98.65 |               0.73 |
>
> Below, we show results for $(\delta_0=0.67, \delta_1=0.67)$.
>
> | Method                                       |    $\epsilon$ |   GPT Judge Rating |
> |:---------------------------------------------|-------:|-------------------:|
> | Watermarked Samples | 0.00 |               0.56 |
> | Qwen/Qwen2.5-3B-Instruct | 76.35 |               0.68 |
> | meta-llama/Llama-3.1-8B-Instruct | 95.61 |               0.71 |
> | Ours-Unigram-Qwen2.5-3B | 97.97 |               0.7  |
> | Ours-Unigram-Llama-3.2-3B | 99.66 |               0.73 |
> | Ours-EXP-Qwen2.5-3B | 99.32 |               0.67 |
> | Ours-KGW-Llama-2-7B | 96.62 |               0.7  |
>
> The results show that our attacks evade detection of this watermark. We will include these results in the revised paper.
>
> > Essential References Not Discussed
>
> Thank you for bringing up these references. We will discuss them in the revised paper.

---

### Official Review · Reviewer_yfhv · 2025-03-14

**Overall Recommendation:** 4

**Summary:**

The paper addresses the vulnerability of Large Language Model (LLM) watermarking methods to adaptive attacks. It argues that existing watermarking robustness tests primarily focus on non-adaptive attackers, which underestimates the risk posed by adversaries with knowledge of the watermarking algorithm. The authors formulate robustness as an objective function and use preference-based optimization to tune adaptive attacks against specific watermarking methods. Their main findings are that adaptive attacks can effectively evade detection across several watermarking methods while maintaining text quality, even with limited computational resources. They also demonstrate that attacks optimized adaptively can remain effective on unseen watermarks in a non-adaptive setting. The key algorithmic idea involves curating preference datasets for fine-tuning adaptive evasion attacks.

## update after rebuttal
Thank you for the author's response. I think this is a good-quality work, and the author has addressed my concerns, so I have decided to raise my score.

**Claims And Evidence:**

The central claim that adaptive attacks are more effective at evading watermarks is supported by the experimental results. The paper provides quantitative data showing that adaptive attacks achieve over 96% evasion success rates against surveyed watermarking methods while preserving text quality.

**Essential References Not Discussed:**

N/A.

**Experimental Designs Or Analyses:**

The experimental design appears sound. The ablation studies cover various settings and hyperparameters. The comparison of adaptive and non-adaptive attacks is well-structured.

**Methods And Evaluation Criteria:**

The proposed methods, including the curation of preference-based datasets and the use of DPO, seem appropriate for tuning watermark evasion attacks. The evaluation criteria include both evasion rate and text quality, which are relevant for assessing the practical impact of the attacks.

**Other Comments Or Suggestions:**

It would be interesting to explore the transferability of adaptive attacks across different types of LLMs (e.g., encoder-decoder models).
The authors could consider investigating the impact of different training objectives for the paraphraser model.

**Other Strengths And Weaknesses:**

Strengths:
- The paper addresses a timely and relevant problem concerning the security of LLM watermarking.
- The paper is clearly written and well-organized.
- The proposed adaptive attack method is effective and scalable.

Weaknesses:
- The paper could benefit from a more in-depth discussion of potential defenses against adaptive attacks, such as adversarial training.
- The reliance on LLM-as-a-judge for text quality assessment is a known limitation. Although this approach is convenient in practice, LLM-as-a-judge may be biased and not necessarily consistent with human judgments.

**Questions For Authors:**

- Could you elaborate on the ethical considerations related to the release of your source code and adaptively tuned paraphrasers, given the potential for misuse in evading existing watermarks?
- Have you explored the possibility of combining your adaptive attacks with other evasion techniques, such as those targeting safety alignment or content filtering, to assess their combined effectiveness? This would give a more comprehensive view of real-world attack scenarios.

**Relation To Broader Scientific Literature:**

The paper builds upon existing literature on LLM watermarking and attack methods. The key contribution lies in highlighting the limitations of non-adaptive robustness testing and proposing a method for adaptive attack optimization, which aligns with the growing awareness of adaptive threats in security research.

**Theoretical Claims:**

This paper primarily focuses on empirical evaluation rather than theoretical contributions.

---

> ### Author Rebuttal · Authors · 2025-03-30
>
> Thank you for your time and valuable suggestions. Please find our answers below.
>
> > The paper could benefit from a more in-depth discussion of potential defenses against adaptive attacks, such as adversarial training.
>
> We appreciate this suggestion. While we focused on the robustness of current watermarking methods to adaptive attacks, we agree that discussing potential defenses would strengthen the paper. In Section 6, we briefly mention the idea of adversarial training to set up a two-player game where both attacker and defenders optimize their strategies, and the goal is to find an Equilibrium. We do not present results on this in our paper. The first challenge would be to develop a watermarking method that can be optimized, following the optimizable attacks we present in this work, which we believe requires substantial effort. This is an interesting direction for future research.
>
> > The reliance on LLM-as-a-judge for text quality assessment is a known limitation. Although this approach is convenient in practice, LLM-as-a-judge may be biased and not necessarily consistent with human judgments.
>
> We thank the reviewer for mentioning this limitation, which we also highlight in our paper. Section 6 acknowledges that LLM-as-a-Judge is an imperfect and noisy metric that may not perfectly align with human judgment. To address this concern, we used multiple evaluation metrics (LLM-Judge, LLM-CoT, LLM-Compare, Mauve, and Perplexity) as described in Appendix A.1, and we have included results with different judge models (both Llama3-8B-Instruct and GPT4o-mini in Appendix A.3). We agree that more work is needed to study the metric's alignment with human judgment as stated in our limitations. However, we believe this work may be outside of the scope of our paper. We will strengthen the discussion by suggesting how future work could incorporate human evaluations to validate our findings.
>
> > It would be interesting to explore the transferability of adaptive attacks across different types of LLMs (e.g., encoder-decoder models).
>
> We agree that it would be interesting to examine different types of LLMs. Our current work focuses on autoregressive decoder-only LLMs. This is the primary setting studied by related work and the most commonly deployed for text generation tasks (e.g., SynthID [A]). Our revised paper will add this as a promising direction for future work. Specifically, we will discuss how the principles of our adaptive attacks might transfer to encoder-decoder architectures like T5 or BART and whether the different generation process might affect the effectiveness of watermarking and our evasion techniques.
>
> > The authors could consider investigating the impact of different training objectives for the paraphraser model.
>
> We appreciate this suggestion. In our current work, we focused on Direct Preference Optimization (DPO) as our training method for the paraphraser. We would kindly ask the reviewer for more information about which specific training objectives they are considering, as this would help us better address their suggestion. In principle, any objective that can optimize for the dual goals of watermark evasion and text quality preservation could be used.
>
> > Could you elaborate on the ethical considerations related to the release of your source code and adaptively tuned paraphrasers, given the potential for misuse in evading existing watermarks?
>
> We carefully considered the ethical implications of releasing our code and models, which we address in our Impact Statement in Section 9. We believe the responsible disclosure of vulnerabilities is essential for improving security systems. Current watermarking deployments are still experimental, and our work highlights vulnerabilities that should be addressed before widespread adoption. By releasing our methods, we enable researchers to test against stronger attacks and build more robust watermarks.
>
> > Have you explored the possibility of combining your adaptive attacks with other evasion techniques, such as those targeting safety alignment or content filtering, to assess their combined effectiveness? This would give a more comprehensive view of real-world attack scenarios.
>
> This is a good point that we also mention as a limitation in our discussion section (Section 6). We focus on robustness of watermarks and do not analyze the interplay of different defenses at the same time.  We agree that examining combined attacks would provide a more comprehensive view of real-world scenarios where multiple defenses might be deployed simultaneously. This is a promising direction for future work and we will discuss this in the revised paper.
>
> ---
> [A] Dathathri, Sumanth, Abigail See, Sumedh Ghaisas, Po-Sen Huang, Rob McAdam, Johannes Welbl, Vandana Bachani et al. "Scalable watermarking for identifying large language model outputs." Nature 634, no. 8035 (2024): 818-823.

---

### Official Review · Reviewer_z2QJ · 2025-03-18

**Overall Recommendation:** 3

**Summary:**

The paper tackles the question of robustness of the generated text from LLMs in the offline setting with adaptive attackers. Their approach is evaluated on a wide variety of LLMs and also weaker paraphasers (LLMs) by considering four watermarking techniques recently introduced in the literature.

Overall: The paper is easy to follow and the approach is relatively straightforward using a preference dataset optimized by using DPO. The setting is novel using adaptive attacks in the offline setting and shows great performance but the underlying approach has limited novelty.

**Claims And Evidence:**

Yes, they are well supported.

**Essential References Not Discussed:**

Check below

**Experimental Designs Or Analyses:**

Yes.

**Methods And Evaluation Criteria:**

Yes, they do.

**Other Comments Or Suggestions:**

See below

**Other Strengths And Weaknesses:**

Pros:

(i) The setting is interesting as it covers the no box,  offline and adaptive settings and still manages to get very impressive results.
(ii) The experiments are extensive and consider a variety of watermarking methods, both non-adaptive and adaptive attacks, and various generative and rephraser LLMs.

Cons:

(a) The technical novelty is limited in that they use standard RL techniques such as DPO to optimize their objective function from a preference dataset.
(b) It is known that scrubbing watermarking is relatively easy and it would have been interesting to have seen work in either spoofing or approaches to thwart these adaptive attacks.

**Questions For Authors:**

(1) How about having a regularization parameter to trade-off quality of text quality and scrubbing?
(2) Can this be utilized for spoofing attacks and if so how could that be implemented?
(3) Also, given that we have this offline attack, is there a way to set this up as game to mitigate for these attacks? It seems relatively easy to scrub. Are there any fundamental results in this space for your setting such as this paper (*).

* Watermarks in the Sand: Impossibility of Strong Watermarking for Generative Models.
 Zhang et al 2023.  https://arxiv.org/abs/2311.04378

**Relation To Broader Scientific Literature:**

Check below

**Theoretical Claims:**

Yes, Algorithm 2.

---

> ### Author Rebuttal · Authors · 2025-03-30
>
> Thank you for your time and effort in providing constructive feedback that will improve our paper. Please find our responses below.
>
> > [..] they use standard RL techniques such as DPO to optimize their objective function from a preference dataset.
>
> Yes, we use standard RL techniques for optimization. However, we view our contributions as follows: (i) Formulating watermark robustness as an objective function, (ii) proposing a framework to optimize this objective adaptively, and (iii) showing the efficacy and efficiency of adaptively optimized paraphrasers. A core finding is that even relatively constrained attackers can evade detection using limited resources.
>
> > How about having a regularization parameter to trade off the quality of text quality and scrubbing?
>
> This is an excellent suggestion. We will introduce a regularization parameter $\beta$ in our objective function (Eq. 2) as follows:
>
> $\underset{\theta_P}{\max} \mathbb{E} \left[\operatorname{Verify}(P_{\theta_P}(x), \tau', m') + \beta Q(P_{\theta_P}(x), x)\right]$
>
> Since we use LoRA adapters, another option that does not require re-training is to scale the weight $\alpha$ of the adapter. When doing so, we measure the following results against EXP using Qwen2.5-3b adaptively tuned against EXP:
>
> | LoRA Strength $\alpha$                   |   Evasion Rate | GPT Judge Rating |
> |:-------------------------|------:|-----------------:|
> | Baseline ($\alpha=0$)   | 69.59 |             0.70 |
> | Ours ($\alpha=0.2$) |  90.20 |             0.72 |
> | Ours ($\alpha=0.4$) |  96.62 |             0.71 |
> | Ours ($\alpha=0.6$) |  97.30 |             0.70 |
> | Ours ($\alpha=0.8$) |  97.97 |             0.70 |
> | Ours ($\alpha=1.0$)   |  96.62 |             0.70 |
>
> We will include these results in our revised paper.
>
> > Can this be utilized for spoofing attacks and if so how could that be implemented?
>
> This is an interesting question and an active area of research. We can share our thoughts on a simple way of how this could be done. First, we would like to point out the difference in settings and objectives: (1) We operate under the no-box setting, but spoofing requires at least some watermarked samples (e.g., black-box access). (2) Spoofing attacks try to maximize the spoofing success rate using a limited number of watermarked samples.
>
> One spoofing attack was proposed by Jovanović et al [1]. In a nutshell, (i) they collect $N$ watermarked samples, (ii) apply a scrubber to create corresponding non-watermarked samples, and then (iii) train a mapper from non-watermarked to watermarked samples. Since our scrubber has substantially higher evasion rates than all existing ones, it could be used as a scrubber in Jovanović's attack to make training the mapper more sample efficient, thus improving the attack. As mentioned before, there are other ways in which our attack could be used, which is an interesting question for follow-up research.
>
> > Also, given that we have this offline attack, is there a way to set this up as game to mitigate for these attacks? It seems relatively easy to scrub. Are there any fundamental results in this space for your setting such as this paper (Watermarks in the Sand).
>
> Again, thank you for this interesting question related to follow-up research. In Section 6, we briefly mention the idea of adversarial training to set up a two-player game where both attackers and defenders optimize their strategies, and the goal is to find Equilibrium. We do not present results on this in our paper. The first challenge would be to develop a watermarking method that can be optimized, following the optimizable attacks we present in this work.
>
> As for fundamental results, there are impossibility results for robustness [2] and provable robustness for watermarks under certain metrics (e.g., token-edit) [3], but we are not aware of fundamental results when treating watermarking as a two-player game.
>
> We hope our responses addressed all your questions and are happy to answer any further questions the reviewer may have.
>
> ---
> [1] Jovanović, Nikola, Robin Staab, and Martin Vechev. "Watermark stealing in large language models." ICML (2024).
>
> [2] Zhang, Hanlin, Benjamin L. Edelman, Danilo Francati, Daniele Venturi, Giuseppe Ateniese, and Boaz Barak. "Watermarks in the sand: Impossibility of strong watermarking for generative models." arXiv preprint arXiv:2311.04378 (2023).
>
> [3] Zhao, Xuandong, et al. "Provable robust watermarking for ai-generated text." arXiv preprint arXiv:2306.17439 (2023).

---

> > ### Comment · Reviewer_z2QJ · 2025-04-09
> >
> > Thanks for the rebuttal and it clarified most of the questions. I will keep my score for now given the simple DPO approach and not strengthening it further by considering other angles such as spoofing.

---

> > > ### Author Response · Authors · 2025-04-09
> > >
> > > Thank you for the response. We are happy to hear that the rebuttal clarified your questions and sincerely appreciate your positive assessment of our work.
> > >
> > > Regarding spoofing: We agree that it is a relevant and interesting security property of watermarks. However, we also highlight that watermarking forgeability is a complex and open research question, and to adequately address it would likely exceed the scope of our paper. That being said, as described in our rebuttal above, we promise to discuss links of our evasion attack to spoofing in the revised paper, such as our attack's potential impact on improving the efficiency of known 'spoofing' attacks (with respect to the number of queries needed to achieve a given spoofing success rate). Please refer to our rebuttal above for more details.
> > >
> > > We hope this improvement addresses your suggestions.

---

### Decision · Program_Chairs · 2025-05-01

**Decision:**

Accept (spotlight poster)

**Comment:**

In view of the fact that many LLM watermarking methods have been proposed, but robustness is tested only against non-adaptive attackers who lack knowledge of the provider’s watermarking method and can find only suboptimal attacks, the authors formulate the robustness of LLM watermarking as an objective function and use preference-based optimization to tune adaptive attacks against the specific watermarking method.

Based on the comments (including post-rebuttal) from FIVE reviewers and the rebuttals from the authors, basically the raised comments have been properly addressed and most reviewers gave positive comments and higher scores.
Although Reviewer Q7gq remained unconvinced that pursuing stronger attacks is the right direction for advancing watermarking, but (s)he was not opposed to the paper being accepted.
Actually, from the viewpoint of AC, it is encouraging to face and solve the challenge of resisting against stronger attacks.

Overall, this submission is a good work and is suggested to be accepted!